# SWIFT4D: ADAPTIVE DIVIDE-AND-CONQUER GAUSSIAN SPLATTING FOR COMPACT AND EFFICIENT RECONSTRUCTION OF DYNAMIC SCENE

**Jiahao Wu,  Rui Peng,  Zhiyan Wang,  Lu Xiao,**
**Luyang Tang**,  **Jinbo Yan**,  **Kaiqiang Xiong**,  **Ronggang Wang**[*]
Guangdong Provincial Key Laboratory of Ultra High Definition Immersive Media Technology
Shenzhen Graduate School, Peking University
`2301212750@stu.pku.edu.cn`,  `rgwang@pkusz.edu.cn`

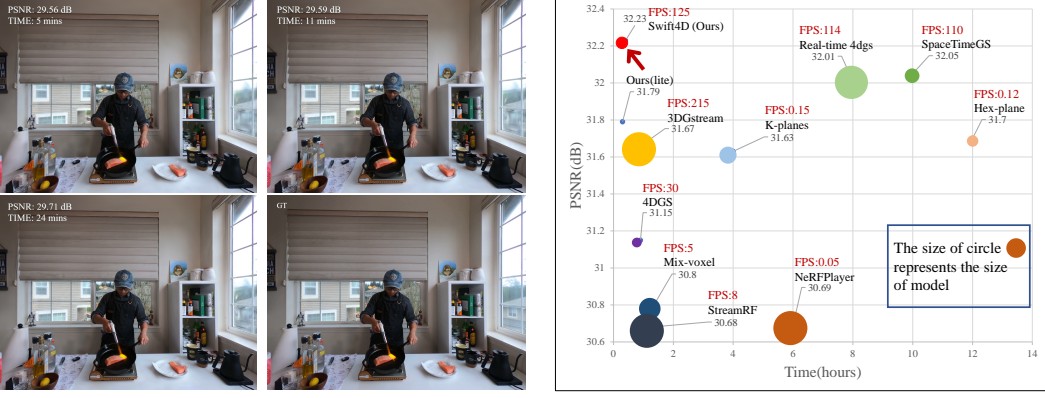

Figure 1: Our method demonstrates high-quality rendering, rapid convergence, and compact storage characteristics. It can achieve competitive result with just 5 minutes of training. Additionally, with increased training iterations, our method excels in handling finer details.

## ABSTRACT

Novel view synthesis has long been a practical but challenging task, although the introduction of numerous methods to solve this problem, even combining advanced representations like 3D Gaussian Splatting, they still struggle to recover high-quality results and often consume too much storage memory and training time. In this paper we propose Swift4D, a divide-and-conquer 3D Gaussian Splatting method that can handle static and dynamic primitives separately, achieving a good trade-off between rendering quality and efficiency, motivated by the fact that most of the scene is the static primitive and does not require additional dynamic properties. Concretely, we focus on modeling dynamic transformations only for the dynamic primitives which benefits both efficiency and quality. We first employ a learnable decomposition strategy to separate the primitives, which relies on an additional parameter to classify primitives as static or dynamic. For the dynamic primitives, we employ a compact multi-resolution 4D Hash mapper to transform these primitives from canonical space into deformation space at each timestamp, and then mix the static and dynamic primitives to produce the final output. This divide-and-conquer method facilitates efficient training and reduces storage redundancy. Our method not only achieves state-of-the-art rendering quality while being 20× faster in training than previous SOTA methods with a minimum storage requirement of only 30MB on real-world datasets.

## 1 INTRODUCTION

Novel view synthesis (NVS) is a crucial task in computer vision and graphics, with significant applications in areas such as augmented reality (AR), virtual reality (VR), and content production.

---
[*]Corresponding author

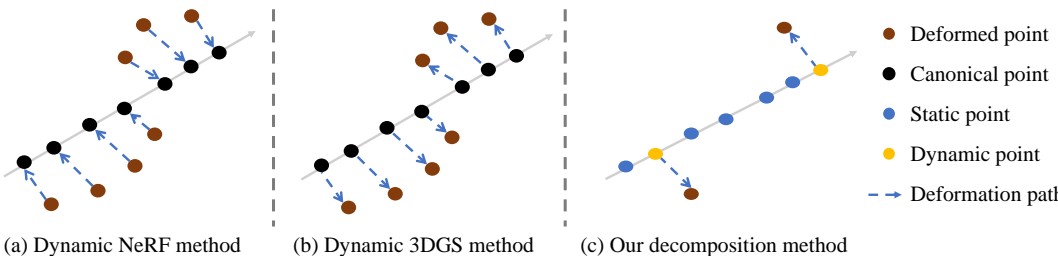

(a) Dynamic NeRF method     (b) Dynamic 3DGS method     (c) Our decomposition method

Figure 2: **Illustration of different dynamic scene rendering methods.** (a) Pumarola et al. (2021); Park et al. (2021) proposes mapping deformation field points to canonical space, a widely adopted practice in NeRF-based methods; (b) Wu et al. (2024); Yang et al. (2024) propose mapping canonical space points to the deformation field; (c) We propose dividing the points in canonical space into dynamic and static, and then mapping only the dynamic points to the deformation space.

The goal of NVS is to render photorealistic images from arbitrary viewpoints using 2D images or video inputs. While recent advancements have achieved considerable success in static scenes, this task becomes particularly challenging when applied to dynamic scenes, where complexities introduced by object motion and temporal changes make accurate rendering significantly difficult.

Current NVS techniques can be broadly classified into two predominant approaches: neural rendering methods, exemplified by Neural Radiance Fields (NeRF) Mildenhall et al. (2021), and point cloud-based rendering techniques, such as 3D Gaussian Splatting Kerbl et al. (2023). NeRFs have recently made significant strides in achieving photorealistic rendering of static scenes, with subsequent works Barron et al. (2021; 2022; 2023); Reiser et al. (2023) further enhancing both quality and speed. Despite these advancements in static scene rendering, NeRFs face significant challenges when extended to dynamic scenes, primarily due to the substantial training time and storage requirements. To overcome these obstacles, various approaches have been proposed. As shown in Fig.2(a), Pumarola et al. (2021) and Park et al. (2021) leverage deformation fields to map deformation space at arbitrary timestamps to canonical space, effectively capturing dynamic scene changes. Li et al. (2021) and Gao et al. (2021) employ scene flow to model the motion trajectories within dynamic environments. Cao & Johnson (2023) and Fridovich-Keil et al. (2023), decompose the 4D spacetime domain into multiple compact planes, thereby improving training and rendering speeds. Although these methods have achieved some degree of success, achieving high-quality real-time rendering remains challenging.

Compared to NeRF, 3DGS offers significant advantages, including real-time rendering and substantially reduced training time. Within the scope of dynamic modeling, several notable methods have emerged. As shown in Fig.2(b), 4DGS Wu et al. (2024), inspired by HexPlane, introduces a neural voxel encoder to model deformation relationships over time. 3DGStream Sun et al. (2024) utilizes a compact Neural Transformation Cache (NTC) to efficiently model the translation and rotation of 3D Gaussians between two adjacent frames. RTGS Yang et al. (2023) treats spacetime as an integrated whole by optimizing a set of 4D primitives, parameterized as anisotropic ellipses that capture both geometry and appearance. Additionally, STGS Li et al. (2024) enhances standard 3D Gaussians with temporal opacity and motion/rotation parameters, effectively capturing both static and dynamic elements to model dynamic deformation.

While these approaches achieve higher-quality results with faster rendering times, they still face challenges related to long training time and heavy storage requirements. One potential limitation in their approach is the uniform treatment of all Gaussian points during the modeling process. However, we observe that static points, such as those in background regions, constitute the majority of the scene. These points exhibit minimal or no deformation and therefore do not require complex dynamic modeling. It is more efficient to partition the scene into static and dynamic points and model each separately. This strategy has the potential to significantly reduce computational overhead and storage requirements. Moreover, as demonstrated by Wang et al. (2023), applying the same modeling technique to both dynamic and static points can cause blurring in dynamic regions due to the influence of static areas, ultimately compromising rendering quality.

In this paper, we introduce Swift4D, a method that simultaneously achieves fast convergence, compact storage, and real-time high-quality rendering. Our approach starts by decomposing Gaussian points into dynamic and static groups based on 2D multi-view images, incorporating an additional parameter $d$ for differentiation. For temporal modeling, we employ a deformation field approach using a compact multi-resolution 4DHash and MLPs as the deformer, which maps dynamic Gaussian points from canonical space to deformation space at arbitrary timestamps. Notably, as

shown in Fig. 2(c), temporal modeling is applied exclusively to the dynamic points, while static points are treated as temporally invariant, significantly reducing computational demands. This reduction in the number of dynamic points enables the 4DHash to concentrate on deformation, leading to faster convergence and improved rendering quality. Finally, we combine the static and dynamic Gaussian points to render the final output. This approach also addresses the issue of blurring caused by static elements interfering with the time-aware multi-resolution 4DHash.

Our method achieves SOTA performance in terms of training and rendering speed, storage efficiency, and rendering quality. Furthermore, our supplement videos ( basketball 1 and 2) demonstrate that our approach remains effective even in scenarios involving large movements. We will release our code and pre-trained models upon acceptance. In summary, the key contributions of our work are:

1) We propose a novel method for decomposing dynamic 3D scenes into dynamic and static components based on 2D images, effectively reducing computational complexity. This method can be seamlessly integrated into existing dynamic approaches as a plug-and-play module to enhance quality.

2) We introduce a compact multi-resolution 4DHash, with a footprint as small as 8MB, to effectively model the spatio-temporal domain. This approach not only enhances rendering quality and accelerates training but also ensures efficient and compact storage.

3) Our method achieves state-of-the-art performance in training and rendering speed, storage, and high-quality output.

## 2 RELATED WORK

**Novel View Synthesis.** In recent years, novel view synthesis has garnered significant attention, leading to numerous breakthroughs. NeRFMildenhall et al. (2021) pioneered this domain by leveraging multi-layer perceptrons (MLPs) combined with volume rendering to model 3D radiance fields, enabling image rendering from arbitrary viewpoints. Subsequent works aimed to enhance efficiency and quality. Methods such as TensorF Chen et al. (2022), DVGO Sun et al. (2022), PlenoxelFridovich-Keil et al. (2022), and Plenoctree Yu et al. (2021) adopt grid-based representations for faster training and rendering. Instant NGP Müller et al. (2022) further accelerates this process with a hash encoder, significantly reducing computation time. Meanwhile, MipNeRF Barron et al. (2021) and MipNeRF360 Barron et al. (2022) propose integrated positional encoding (IPE) to model conical frustums, effectively mitigating aliasing issues. More recently, 3DGS Kerbl et al. (2023) introduced a novel point-based rendering paradigm for novel view synthesis, achieving real-time rendering with high quality. This has spurred additional advancements, including Mipsplatting Yu et al. (2024a) for anti-aliasing, 2DGS Huang et al. (2024a) for improved mesh extraction, and ScaffoldGS Lu et al. (2024) for large-scale scene rendering.

**Novel View Synthesis for dynamic scene.** Li et al. (2021); Lin et al. (2024); Kratimenos et al. (2023) attempt to directly model the trajectories of moving points across the scene, but they continue to encounter challenges related to storage. Pumarola et al. (2021); Park et al. (2021); Wu et al. (2024); Yang et al. (2024) try to build a consistent canonical space across each time step and then employ a deformer, mainly MLP-based and Muti-plane-based, to map this canonical space to deformation spaces at each timestamp. Huang et al. (2024b) focuses on monocular dynamic inputs, leveraging sparse control points to reconstruct scene dynamics with exceptionally high FPS. Lin et al. (2024) employs Fourier series and polynomial fitting to model the motion of Gaussian points, enabling dynamic reconstruction. K-planes Fridovich-Keil et al. (2023) and Hexplane Cao & Johnson (2023) employ an explicit structural representation of the 6D light field rather than modeling underlying motions. Representing the deformer using MLPs or low-rank planes can reduce storage requirements, but it often results in slower training and limited capacity for capturing complex deformations.

Recently, He et al. (2024); Yan et al. attempt to separate dynamic and static Gaussian points to improve rendering quality and introduced external models to segment foreground and background areas. While these efforts have explored this direction, the resulting output quality remains suboptimal. Liang et al. (2023) employs adaptive dynamic-static separation, which differs from our explicit separation approach.

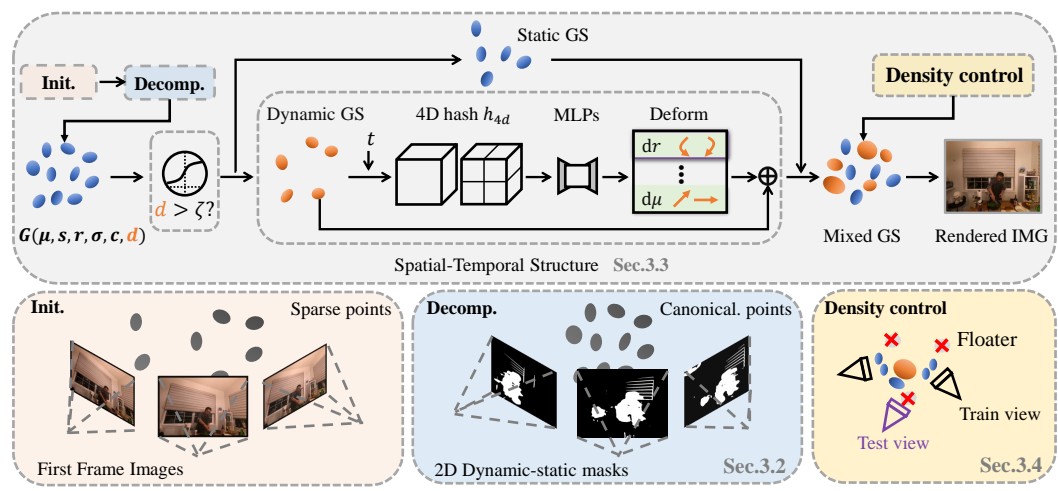

Figure 3: **Pipeline of our Swift4D.** First, we use the first frame images to obtain a well-initialized canonical point cloud. Then, we train the dynamic parameter $d$ according to the method described in Sec.3.2. Based on $d$, the point cloud is divided into dynamic and static categories. Dynamic points undergo deformation using a spatio-temporal structure, as discussed in Sec.3.3. Finally, the deformed dynamic points are mixed with static points for rendering.

## 3 METHOD

Our main approach aims to achieve faster training speeds and higher quality rendering results through the decomposition of dynamic and static elements. Based on this insight, we designed our pipeline, as illustrated in Fig. 3. In this section, we will provide a detailed analysis of each module in the pipeline. The preliminary concepts of 3D Gaussian Splatting are briefly introduced in Sec. 3.1. We initially train the canonical space Gaussians using the first-frame images and then optimize the dynamic parameter $d$ of each Gaussian point based on the 2D dynamic-static pixel masks from different viewpoints, as discussed in Sec.3.2. In the following stage, as outlined in Sec. 3.3, we freeze the training of dynamic parameter $d$ and proceed to jointly optimize the remaining parameter of the Gaussian points alongside the swift spatio-temporal structure. Furthermore, our pruning strategies are thoroughly described in Sec. 3.4, while Sec. 3.5 provides an in-depth discussion of the optimization process.

### 3.1 GAUSSIAN SPLATTING PRELIMINARY

3DGSKerbl et al. (2023) uses 3D Gaussian points as its rendering primitives. These 3D Gaussian points have the following parameter: mean $\mu$, covariance matrix $\Sigma$, opacity $\sigma$, and view-dependent color $c$. A 3D Gaussian point is mathematically defined as:

$$G(x) = e^{-\frac{1}{2}(x-\mu)^T \Sigma^{-1}(x-\mu)} \tag{1}$$

In the next rendering phase, the 3D mean $\mu$ is directly projected onto the plane as a 2D mean $\mu^{2D}$, while the 3D covariance matrix is transformed into a 2D covariance matrix using the following formula: $\Sigma' = (JW\Sigma W^T J^T)$, where $W$ and $J$ denote the viewing transformation and the Jacobian of the affine approximation of the perspective projection transformation, respectively. Finally, the color of each pixel can be calculated using the following formula:

$$C(x) = \sum_{i \in \mathcal{N}(\mathbf{x})} \mathbf{c}_i \alpha_i(x) \prod_{j=1}^{i-1}(1 - \alpha_j(x)) \text{ where } \alpha_i(x) = \sigma_i \exp\left(-\frac{1}{2}(x - \mu_i^{2D})^T \Sigma'^{-1}(x - \mu_i^{2D})\right). \tag{2}$$

Where $N$ is the number of Gaussian points that intersect with the pixel $x \in \mathbb{R}^2$. In the actual implementation, the covariance matrix $\Sigma$ is typically decomposed into rotation $q$ and scaling $s$. The color $c$ is represented by a spherical harmonics (SH) function. Therefore, a Gaussian point can be represented as $G\{\mu, q, s, \sigma, c\}$.

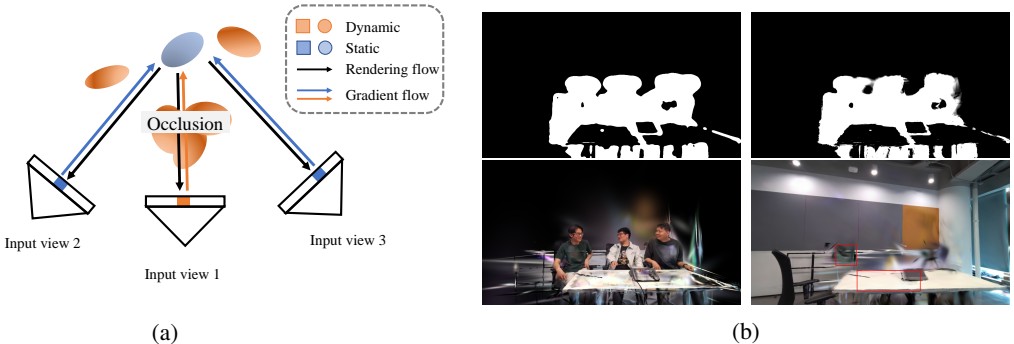

(a)                    (b)

Figure 4: (a) Diagram illustrating dynamic parameter $d$ optimization. Even when static points (blue) are occluded by dynamic points (orange) from View 1, they can still be correctly optimized from View 2 and 3. (b) shows the result of decomposition. From top left to bottom right, the order is GT mask, dynamic parameter rendered image, dynamic point and static point rendering results.

## 3.2 EFFICIENT DYNAMIC AND STATIC DECOMPOSITION

In this section, we introduce our dynamic-static decomposition method for eliminating redundant computations for static Gaussian points. The time-varying motion model is applied solely to the dynamic components, leaving the static elements unchanged. This approach leads to faster convergence and enhanced rendering quality. Specifically, we introduce a learnable dynamic parameter $d$ (Initialized to 0.) within the Gaussian points to quantify the the dynamic level of each points. A higher $d$ corresponds to more pronounced motion, indicating that the point is likely dynamic. We first compute a 2D dynamic-static pixel mask $D(x)$ from the training videos to distinguish dynamic and static pixels, as shown in Eq. 3, which serves as the supervision signal.

$$D(x) = \begin{cases} 1 & S(x) >= \gamma \\ 0 & S(x) < \gamma \end{cases} \quad \text{where} \quad S(x) = \sqrt{\frac{1}{T}\sum_{t=1}^{T}(C(x,t) - \frac{1}{T}\sum_{t=1}^{T}C(x,t))^2} \quad (3)$$

where C(x, t) represents the pixel intensity of in $t$ th frame at location $x \in \mathbb{R}^2$. $S(x)$ is the temporal standard deviation (std) for each pixel $x$ across the entire time duration $T$. Subsequently, a threshold of $\gamma = 0.02$ is applied to binarize $S(x)$, generating a pixel mask $D(x)$. Pixels with $S(x)$ greater than or equal to $\gamma$ are classified as dynamic, while those below are considered static.

Based on the concept: During backpropagation, Gaussian points intersecting dynamic pixels should receive a positive gradient, while those intersecting static pixels should receive a negative gradient, with the gradient gradually weakening with distance and occlusion. Our decomposition design is illustrated in Fig. 4(a). We use the $\alpha$ composition for parameter $d$ with Sigmoid function to render a dynamic value $\hat{D}(x)$ at location x, as Eq. 4 shows.

$$\hat{D}(x) = \text{Sigmoid}(\sum_{i=1} d_i\alpha_i(x)\prod_{j=1}^{i-1}(1 - \alpha_j(x))) \quad (4)$$

By applying the **sigmoid** function, we optimize the dynamic parameter $d$ to span $(-\infty, +\infty)$, enabling finer differentiation of dynamic degrees. This approach converts our dynamic-static decomposition into a binary classification problem. Consequently, optimizing the dynamic value for each Gaussian point can be accomplished by minimizing the binary cross-entropy loss:

$$\mathcal{L}_d = \mathbb{E}_x[-D(x)\log(\hat{D}(x)) - (1 - D(x))\log(1 - \hat{D}(x))]. \quad (5)$$

From the equation above, we effectively optimize the dynamic parameter $d$ for the Gaussian points. The entire optimization process is highly efficient, typically concluding within 1 minute. Ultimately, Gaussian points with dynamic parameter greater than the dynamic threshold $\zeta = 7.0$ are classified as dynamic; otherwise, they are classified as static. An ablation study on $\zeta$ is presented in Fig. 7 and Sec. 5, demonstrating that our decomposition method is robust to the choice of $\zeta$.

Notably, our method adapts well to occlusion. In Fig. 4(a), while dynamic pixels (orange) from View 1 incorrectly assign positive dynamic values to static Gaussian points (blue), other views like View

2 and 3 assign larger negative values, ensuring correct classification. Further details are provided in the Supplementary material Sec. A.2.

### 3.3 SPATIO-TEMPORAL STRUCTURE

We introduce our proposed efficient spatio-temporal structure encoder, the 4D multi-resolution hash $h_{4d}$, and the deformation decoder MLPs, used to predict the deformation of each dynamic Gaussian.

**4D Multi-resolution hash encoder.** Inspired by INGP Müller et al. (2022), we propose utilizing a 4D multi-resolution hash $h_{4d}$ for encoding to effectively model the temporal information of dynamic Gaussians by normalizing the point cloud into the hash grid range. As described in INGP, voxel grid at each resolution is mapped to a hash table that stores $F$-dimensional learnable feature vectors. For a given 4D dynamic Gaussian $(\mu, t) \in \mathbb{R}^4$, its hash encoding at resolution $l$, denoted as $h_{4d}(\mu, t; l) \in \mathbb{R}^F$, is computed through linear interpolation of the feature vectors associated with corners of the surrounding grid. Consequently, its multi-resolution hash encoding features are as follows:

$$f_h = [h_{4d}(\mu, t; 0), h_{4d}(\mu, t; 1)...h_{4d}(\mu, t; L-1)] \in \mathbb{R}^{LF}, \tag{6}$$

where $L$ denotes the number of resolution levels, typically set to 16. Following this, a small MLP $\phi_d$ combines all features to produce $f_d = \phi_d(f_h)$. Using the 4D hash $h_{4d}$ as an encoder offers several advantages: compactness, $O(1)$ query complexity, and the multi-resolution approach effectively integrates global and local information.

However, while 4D Hash offers $O(1)$ query complexity, its hashing characteristics make encoding the temporal information of an entire scene both challenging and storage-intensive. Fortunately, our proposed decomposition method focuses on encoding the temporal information of dynamic points only, reducing the need for a larger hashing space and simplifying the modeling of the scene's temporal domain. This approach enables us to retain the fast access speed of 4D Hash while minimizing storage requirements.

**Multi-head Gaussian Deformation Decoder.** Once all features of dynamic Gaussian points are encoded, we can compute any required variables using a multi-head Gaussian deformation decoder MLPs = $\{\phi_\mu, \phi_s, \phi_q, \phi_\sigma, \phi_{sh}\}$:

$$\mathrm{d}\mu, \mathrm{d}s, \mathrm{d}q, \mathrm{d}\sigma, \mathrm{d}sh = \mathrm{MLPs}(f_d) \tag{7}$$

Here, $\mathrm{d}\mu, \mathrm{d}s, \mathrm{d}q, \mathrm{d}\sigma, \mathrm{d}sh$ represent the deformation intensity of the mean, scaling, rotation, opacity, and color of the Gaussian point at time $t$. Therefore, the deformed parameters of dynamic Gaussian $G_d$ can be expressed as:

$$(\mu', r', q', \sigma', sh') = (\mu + \mathrm{d}\mu, r + \mathrm{d}r, q + \mathrm{d}q, \sigma + \mathrm{d}\sigma, sh + \mathrm{d}sh) \tag{8}$$

where ( $\mu', r', q', \sigma', sh'$ ) represent the new parameters of the dynamic Gaussian at time $t$. For static Gaussian elements $G_s$, they are directly combined with the deformed dynamic Gaussian elements $G_d$ to render the final rendered image $I_t$.

### 3.4 DENSITY CONTROL

In the original 3DGS, the opacity of all points is regularly reduced, and Gaussian points with low transparency are clipped during the pruning stage. However, this approach is not appropriate for our method as it results in excessive coupling between the canonical space and the deformation space. Therefore, we eliminate the reset opacity operation. Inspired by previous works Niemeyer et al. (2024); Deng et al. (2024); Fan et al. (2023), which focus on the compact representation of static scenes by pruning redundant Gaussians based on spatial attributes such as transparency and volume. We adopt a novel approach to pruning floaters across canonical and deformation space: Temporal Importance Pruning, as shown in Fig. 3. This involves calculating the importance of each Gaussian point to each training viewpoint at every timestamp. Gaussians with importance below a certain threshold can be clipped, effectively reducing floater issues. For a Gaussian point $g_i$, the importance $w_i$ is calculated as follows:

$$w_i = \max_{I \in \mathcal{I}, x \in I, t \in T} (\alpha_i(x|t) \prod_{j=1}^{i-1} (1 - \alpha_j(x|t))) \tag{9}$$

Table 1: **Quality comparison on the N3DV dataset.** The best and the second best results are denoted by red and blue. [1] online method.

| Method | PSNR ↑ | DSSIM ↓ | LPIPS ↓ | Time ↓ | Size(MB) ↓ | FPS ↑ |
|--------|--------|---------|---------|--------|------------|-------|
| DyNeRF Li et al. (2022b) | 29.58 | 0.020 | 0.099 | 1300.0 hours | 30 | 0.02 |
| NeRFPlayer Song et al. (2023) | 30.69 | - | 0.111 | 6.0 hours | 5100 | 0.05 |
| HexPlane Cao & Johnson (2023) | 31.70 | 0.014 | 0.075 | 12.0 hours | 240 | 0.21 |
| K-Planes Fridovich-Keil et al. (2023) | 31.63 | 0.018 | - | 5.0 hours | 300 | 0.15 |
| 4DGS Wu et al. (2024) | 31.02 | 0.030 | 0.150 | 50 mins | 90 | 30 |
| 3DGStream [1] Sun et al. (2024) | 31.67 | - | - | 60 mins | 2340 | 215 |
| SpaceTimeGS Li et al. (2024) | 32.05 | 0.014 | 0.044 | 10.0 hours | 200 | 110 |
| Real-Time4DGS Yang et al. (2023) | 32.01 | 0.014 | 0.055 | 9.0 hours | > 1000 | 114 |
| **Swift4DLite(Ours)** | 31.79 | 0.017 | 0.072 | 20 mins | 30 | 128 |
| **Swift4D(Ours)** | 32.23 | 0.014 | 0.043 | 25 mins | 120 | 125 |

Table 2: **Quantitative comparison on the MeetRoom dataset.** PSNR is averaged across all frames, while training time and storage requirements accumulate over the entire sequence. [1] online method.

| Method | PSNR ↑ | Time(hours) ↓ | Size(MB) ↓ |
|--------|--------|---------------|------------|
| PlenoxelFridovich-Keil et al. (2022) | 27.15 | 70 | 304500 |
| I-NGPMüller et al. (2022) | 28.10 | 5.5 | 14460 |
| 3DGSKerbl et al. (2023) | 31.31 | 13 | 6330 |
| StreamRF [1] Li et al. (2022a) | 26.72 | 0.85 | 2700 |
| 3DGStream [1] Sun et al. (2024) | 30.79 | 0.6 | 1230 |
| **Swift4D(Ours)** | 32.05 | 0.3 | 40 |

Here, $\mathcal{I}$ represents the images from all training views, $\alpha_i(x|t)$ is the value of $\alpha_i(x)$ at time $t$ in Eq. 2, $T$ represents the set of query times. We prune Gaussians when their importance satisfies $w_i < 0.02$. As illustrated in Fig. 9, this method effectively eliminates artifacts that are suspended in the air and were not captured by the training views. For the cloning and splitting of Gaussians, we adhere to the procedures of 3DGS, with the child Gaussians inheriting the dynamic properties of their parent Gaussians.

### 3.5 OPTIMIZATION PIPELINE

We start by initializing the SfM Schonberger & Frahm (2016) point cloud using the first frames, then train on the first-frame images for 5000 iterations to establish a well-defined canonical space. Next, training the dynamic attributes of each Gaussian point within the canonical space takes about 1 minute, followed by training the spatio-temporal structure. Consistent with the principles of 3DGS, our loss function remains simple, without additional terms:

$$\mathcal{L}_{rec} = (1 - \lambda_1)\mathcal{L}_1 + \lambda_1\mathcal{L}_{SSIM}$$

## 4 EXPERIMENT

In this section, we provide details of our implementation and datasets in Sec. 4.1 and 4.2, respectively. A thorough analysis of the experimental results is presented in Sec. 4.3, while Sec. 4.4 covers the ablation experiments for our method. The results show that our approach achieves sota performance in terms of training speed, storage efficiency, and rendering quality.

### 4.1 IMPLEMENTATION DETAILS

We initialize with point clouds generated by Colmap, followed by constructing our canonical space using the first frames from all training viewpoints, trained for 5000 epochs. Next, we train the dynamic parameter $d$ of Gaussian points using the Adam optimizer Kingma & Ba (2014) with a learning rate of 0.05. This training spans 3000 epochs and completes in under 1 minute. Finally, we train our spatio-temporal structure for approximately 14000 epochs, utilizing settings for the 4D Hash table similar to those in InstantNGP Müller et al. (2022). We use the Adam optimizer with an initial learning rate of 0.002, which exponentially decays to 0.00002 over the course of training.

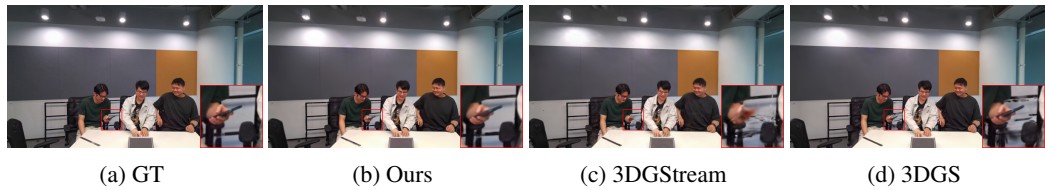

| (a) GT | (b) Ours | (c) 3DGStream | (d) 3DGS |

Figure 5: Qualitative result on the *discussion*.

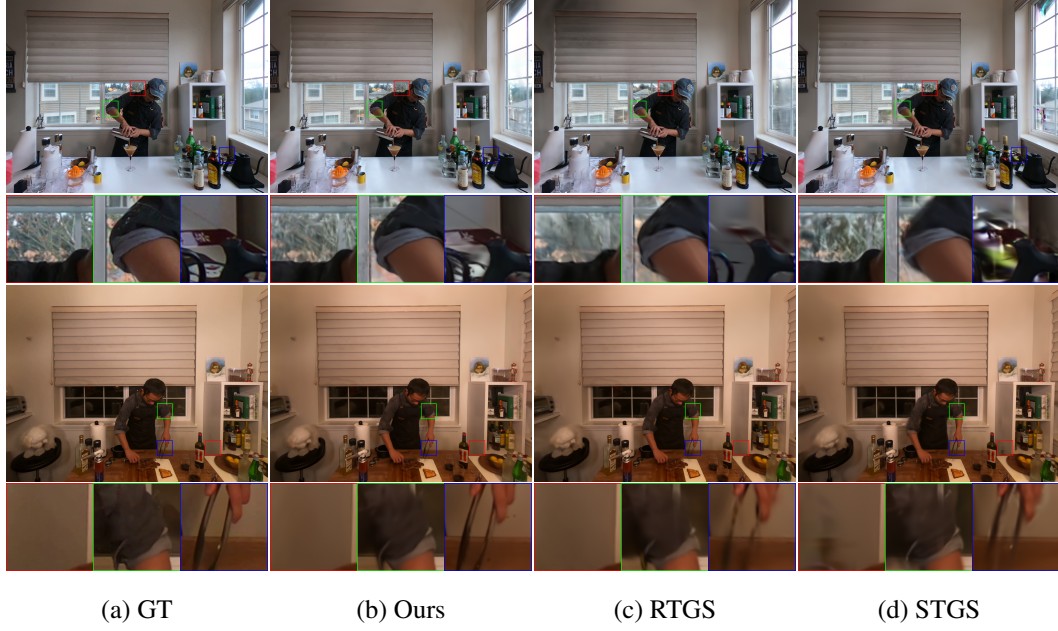

| (a) GT | (b) Ours | (c) RTGS | (d) STGS |

Figure 6: Qualitative result on *coffee martini* and *cut beef*. It can be observed that our method achieves higher-quality modeling in both dynamic and static regions.

Lite refers to a lite-version model with a hash table size set to $2^{15}$ and $\lambda_1 = 0$. All experiments were conducted on an NVIDIA RTX 3090 GPU.

## 4.2 DATASET

**The N3DV dataset Li et al. (2022b)** is captured using a multi-view system with 18-21 cameras, recording dynamic scenes at a resolution of $2704 \times 2028$ and 30 FPS. It includes various complex scenarios such as fire, reflections, and new objects. Following prior works Li et al. (2022b); Cao & Johnson (2023); Wu et al. (2024); Yang et al. (2023); Li et al. (2024), we downsampled the videos by a factor of two and used the same training and testing data splits as established by them.

**The Meet Room dataset Li et al. (2022a)** is captured using a multi-view system with 13 cameras, recording dynamic scenes at a resolution of $1280 \times 720$ and 30 FPS. Following prior works Sun et al. (2024); Li et al. (2022a), we used 12 views for training and reserved 1 view for testing.

**The Basketball court dataset VRU (2024)** is captured using a multi-view system with 34 cameras, recording dynamic scenes at a resolution of $1920 \times 1080$ and 25 FPS. This dataset encompasses a large scene with many complex situations, including bouncing, fast motion, occlusion, and transient objects, making it highly challenging.

## 4.3 EVALUATION

For the Meetroom and Basketball court dataset, we follow the processing approach from Sun et al. (2024), using COLMAP Schonberger & Frahm (2016) to estimate the camera pose of the first frame as the global pose. For the N3DV dataset, we adopt the processing approach from Wu et al. (2024). We evaluate the methods using three metrics across all 300 frames: 1) Average PSNR, DSSIM, and LPIPS Zhang et al. (2018) scores for the test views; 2) Total training time and FPS; 3) Model size .

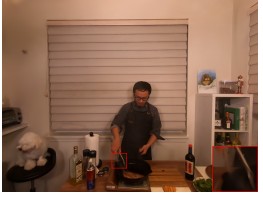 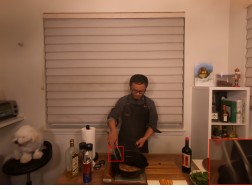 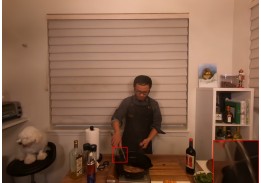 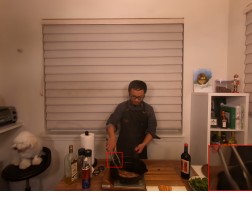

(a) Lite     (b) Muti-plane encoder    (c) W/O decomp    (d) Full(Ours)

Figure 8: Some ablation experiments results on *sear steak*.

Tab. 1 and Fig. 6 respectively present the quantitative and qualitative evaluations of various methods on the N3DV video dataset. As shown in Tab. 1, our approach not only significantly surpasses previous methods in rendering quality but also achieves speeds at least 20 times faster compared to methods achieving similar rendering quality Yang et al. (2023); Li et al. (2024). It can be seen in Fig. 6 that our method not only achieves higher-quality modeling of static regions, such as the plate in the bottom right corner and the background outside the window in the *coffee martini*, but also provides more detailed modeling of moving regions, such as the arms. As for the MeetRoom dataset results, shown in Fig. 5 and Tab. 2, our method achieves state-of-the-art performance in rendering quality, training time, and storage efficiency. Particularly noteworthy is the storage efficiency, as 3DGStream requires 30 times more storage compared to our approach. The results of training on the basketball court dataset are presented in Fig. 10, Appendix Tab. 5, and the supplementary videos (basketball 1 and 2), showcasing our method's ability to handle highly complex dynamic scenes. Our decomposition technique effectively separates all athletes from the scene, illustrating the model's strong adaptability to occlusion, as discussed in Sec. 3.2.

Table 3: Quantitative results comparison for *sear steak* and *flame steak* includes average PSNR and time metrics over 300 frames of the test view.

| Method | PSNR ↑ | Time(mins) ↓ |
|---|---|---|
| Lite | 33.31 | 20 |
| Muti-planes | 33.48 | 28 |
| W/o decomp. | 32.68 | 35 |
| Full | 33.83 | 25 |

## 4.4 ABLATION AND ANALYSIS

**Dynamic and static decomposition.** To validate the effectiveness of our dynamic-static decomposition method, we conducted experiments on the *sear steak* and *flame steak*. As illustrated in Fig. 8(c) and Tab. 3, treating all points as dynamic led to increased computation time, significantly reduced rendering quality, and introduced blurring in areas with large motion amplitudes.

**Muti-plane encoder.** There are three commonly used choices for encoder selection: an implicit MLP Gao et al. (2021), multi-planes Cao & Johnson (2023), and a hash table Müller et al. (2022). In this study, we delve into employing the multi-plane approach to replace the 4D Hash as the encoder in our method. The subjective and objective experimental results, shown in Fig. 8(b) and Tab. 3, indicate that while it slightly lags behind the hash table in terms of rendering speed and quality, it still outperforms methods that do not apply dynamic-static decomposition Wu et al. (2024).

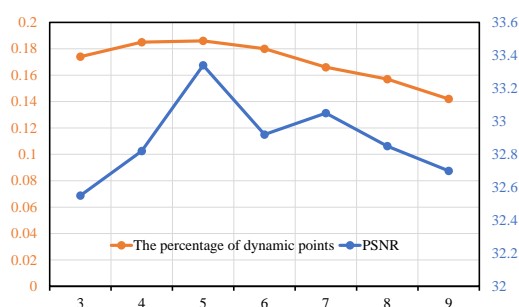

Figure 7: **Distribution of dynamic points counts and PSNR at different thresholds.**

**Temporal importance pruning.** As shown in Fig. 9, (a) and (b) exhibit severe artifacts. In contrast, images rendered with our pruning strategy, (c), appear much cleaner.

## 5 DISCUSSION

**Incomplete decomposition of dynamic and static points.** Although we employ the pixel level supervisor, it fails to fully decouple dynamic and static points. This can lead to two issues: Gaussian

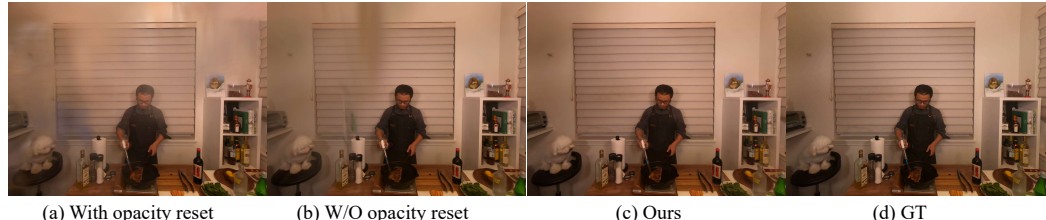

|     |     |     |     |
|-----|-----|-----|-----|
| (a) With opacity reset | (b) W/O opacity reset | (c) Ours | (d) GT |

Figure 9: **Importance Pruning Ablation Experiments:** (a), (b), (c), and (d) show the rendered results of the our model with opacity reset every 3000 iterations, without opacity reset, with our importance pruning method, and the ground truth, respectively.

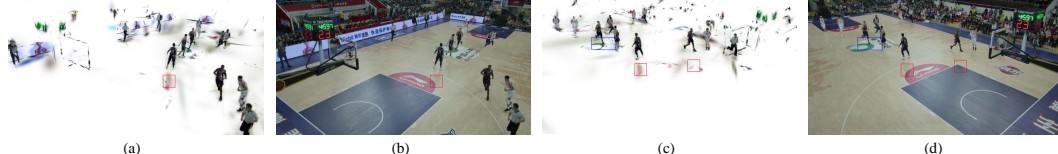

|     |     |     |     |
|-----|-----|-----|-----|
| (a) | (b) | (c) | (d) |

Figure 10: **Basketball court dataset experiment.** (a) and (c) are dynamic point renderings, while (b) and (d) are GT. The black floaters are actually Gaussian points from the dynamic background.

points in textureless regions of small moving objects may be mistaken for static, while static objects may be identified as dynamic due to interference from nearby moving objects.

In the first situation, as shown in Fig. 4(b), certain textureless areas, like clothing and the table, are mistakenly identified as static, despite being dynamic. In fact, this proves beneficial in Tab. 3. If the entire table were labeled as dynamic, the increased dynamic points would lower rendering quality (W/o decomp). By recognizing only the edges as dynamic, where pixel changes are significant, the method reduces the number of dynamic points and enhances rendering quality (Ours).

In the second situation, as illustrated in Fig. 10, some Gaussians in static areas are classified as dynamic due to the shadows or movements of the basketball players passing through these regions. Therefore, it is reasonable to identify these areas as dynamic. Classifying these points as static will impact the visual experience, as static points can hardly model dynamic areas.

**The selection of the dynamic threshold.** As shown in Fig. 7, experiments with the "cook spinach" scene revealed that varying the dynamic threshold $\zeta$ from 3 to 9 did not significantly affect PSNR or the percentage of dynamic points, demonstrating the robustness of our method. To ensure consistency, we set the threshold to 7.

**The training time.** Due to the large dataset (nearly 6000 images), we load images during training, which empirically wastes around $40\%$ of the training time on disk I/O. Eliminating this overhead could reduce training time to 10 minutes while maintaining high-quality 4D scene reconstruction.

## 6 CONCLUSION

In this paper, we introduce Swift4D, which achieves fast convergence, compact storage, and high-quality real-time rendering capabilities within the field of 4D reconstruction. The core innovation of our method lies in the introduction of a dynamic-static decomposition technique, which can be applied to most existing dynamic scene reconstruction methods, enhancing quality and accelerating convergence. Additionally, we introduce a 4D Hash encoder and a multi-head decoder as our spatio-temporal structure, allowing for faster and more efficient temporal modeling of dynamic points. Finally, to prevent severe coupling between the canonical and deformation fields, we propose a novel temporal pruning method that effectively removes floaters in the scene. Our proposed method delivers competitive results in just 5 minutes, and we hope it can offer new insights for applications struggling with training efficiency.

**Limitation:** Similar to previous work Sun et al. (2024); Li et al. (2024), our mrthod focuses on multi-view scenes and currently does not support monocular datasets for dynamic scene reconstruction. Additionally, our method focuses on scene reconstruction and does not include human reconstruction Wu et al. (2020); Cheng et al. (2023).

## 7 ACKNOWLEDGEMENTS

This work is financially supported by Guangdong Provincial Key Laboratory of Ultra High Definition Immersive Media Technology(Grant No. 2024B1212010006), National Natural Science Foundation of China U21B2012, Shenzhen Science and Technology Program-Shenzhen Cultivation of Excellent Scientific and Technological Innovation Talents project(Grant No. RCJC20200714114435057), this work is also financially supported for Outstanding Talents Training Fund in Shenzhen.

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

# A  APPENDIX

In the supplementary materials, we will provide more details. In Sec.A.1, we provide detailed setting s about our method. In Sec.A.2, we describe our dynamic-static decomposition method in detail. In Sec.A.3, we present additional experimental results.

## A.1  IMPLEMENT DETAILS

**Three-stage method.** In the first stage, we train the Gaussian points initialized by SfMSchonberger & Frahm (2016) using the first frame images from each viewpoint. The goal of this stage is to obtain a canonical space. In the second stage, we train the dynamic parameter $d$ of each Gaussian point according to the method proposed in Sec. 3.2. In the third stage, we jointly train the attributes of the Gaussian points and the spatio-temporal structure.

**MLPs as feature decoders.** As shown in Fig.13. we use five shallow MLPs as decoders for the mean, opacity, color, rotation, and scaling, respectively. The outputs are directly added to the attributes of the Gaussian points in the canonical space, and then passed through the corresponding activation functions to obtain the attributes at time t.

**Lite version.** We empirically found that removing the SSIM loss, while slightly degrading rendering quality, offers the advantage of reducing the number of Gaussian points by 2-3 times (approximately 200,000). Therefore, we removed the SSIM loss and set the hash table size to $2^{15}$ in the Lite version. This ensures a significant reduction in model size without severely impacting rendering quality. The models rendered in the Lite version average only 30MB in size, with the Gaussian point cloud being 22MB and the hash table 8MB, making it the smallest dynamic model to date (including the NeRF series).

## A.2  DYNAMIC - STATIC DECOMPOSITION

We precompute the temporal variance $S_i^2(x)$ for each pixel $x \in \mathbf{R}$ to generate the variance map $V_i$ for each viewpoint $i$. To reduce noise, we smooth $V_i$ using a Gaussian filter with a 31x31 kernel. Each pixel's variance $S_i^2(x)$ is then binarized into $D_i(x)$ using a threshold $\gamma$, providing pixel-level supervision.

Initially, the dynamic parameter $d$ of each Gaussian point is set to 0, resulting in a dynamic value of $\hat{D}_i(x) = 0.5$ for each pixel. When the cross-entropy loss $\mathcal{L}_d$ is employed as the loss function, the Gaussian points that intersect with the dynamic pixel $\hat{D}_i(x)$ will receive a positive gradient, leading to the dynamic parameter $d$ expanding towards $+\infty$. Conversely, when Gaussian points intersect with static pixels, the dynamic parameter $d$ will expand towards $-\infty$. Due to the properties of the **Sigmoid** function, the dynamic parameter can extend infinitely towards both $-\infty$ and $+\infty$, allowing us to better distinguish between dynamic and static points.

When a Gaussian point intersects both dynamic and static pixels (e.g., in the presence of occlusion), it will receive two opposing gradient values. If the positive gradient is larger, its dynamic value will be greater than 0, classifying it as a dynamic point. Conversely, if the negative gradient dominates, it will be classified as a static point.

Table 4: **Per-scenes results on the NV3D dataset.** The best and the second best results are denoted by red and blue.

| Method | Coffee Martini | Spinach | Cut Beef | Flame Salmon | Flame Steak | Sear Steak | Mean |
|---|---|---|---|---|---|---|---|
| MixVoxels | 29.36 | 31.61 | 31.30 | 29.92 | 31.21 | 31.43 | 30.80 |
| NeRFPlayer | 31.53 | 30.56 | 29.35 | 31.65 | 31.93 | 29.13 | 30.69 |
| HexPlane | – | 32.04 | 32.55 | 29.47 | 32.08 | 32.39 | 31.70 |
| K-Planes | 29.99 | 32.60 | 31.82 | 30.44 | 32.38 | 32.52 | 31.63 |
| 4DGS | 27.34 | 32.46 | 32.90 | 29.20 | 32.51 | 32.49 | 31.15 |
| 3DGStream | 27.75 | 33.31 | 33.21 | 28.42 | 34.30 | 33.01 | 31.67 |
| SpaceTimeGS | 28.61 | 33.18 | 33.52 | 29.48 | 33.64 | 33.89 | 32.05 |
| Real-Time4DGS | 28.33 | 32.93 | 33.85 | 29.38 | 34.03 | 33.51 | 32.01 |
| **Swift4DLite(Ours)** | 28.84 | 32.57 | 32.82 | 29.92 | 33.13 | 33.48 | 31.79 |
| **Swift4D(Ours)** | 29.13 | 33.05 | 33.80 | 29.75 | 33.67 | 33.98 | 32.23 |

Finally, we provide the formula for calculating the gradient received by each Gaussian point. Based on this formula, the CUDA code can be easily written. Assuming we need to compute the dynamic value gradient of Gaussian point $g$, the equation as following: $\frac{\partial \mathcal{L}_d}{\partial d_g}$. Due to *autograd* , we have known $grad_1$:

$$grad_1 = \frac{\partial \mathcal{L}_d}{\partial(\sum_{i=1} d_i \alpha'_i \prod_{j=1}^{i-1}(1 - \alpha'_j))} \tag{10}$$

we only need to compute:

$$grad_2 = \frac{\partial(\sum_{i=1} d_i \alpha'_i \prod_{j=1}^{i-1}(1 - \alpha'_j))}{\partial d_g} = (\alpha'_g \prod_{j=1}^{g-1}(1 - \alpha'_j)) \tag{11}$$

So, the final formula is as follows:

$$\frac{\partial \mathcal{L}_d}{\partial d_g} = \frac{\partial \mathcal{L}_d}{\partial(\sum_{i=1} d_i \alpha'_i \prod_{j=1}^{i-1}(1 - \alpha'_j))} * \frac{\partial(\sum_{i=1} d_i \alpha'_i \prod_{j=1}^{i-1}(1 - \alpha'_j))}{\partial d_g} \tag{12}$$

$$= \frac{\partial \mathcal{L}_d}{\partial(\sum_{i=1} d_i \alpha'_i \prod_{j=1}^{i-1}(1 - \alpha'_j))} * (\alpha'_g \prod_{j=1}^{g-1}(1 - \alpha'_j)) \tag{13}$$

From this formula, it can be seen that the gradient of the dynamic value is related to occlusion, self-opacity, and the distance to the camera plane, which is very reasonable.

### A.3 MORE RESULTS

Fig.11 shows the rendering results from new viewpoints at different iteration of training. It can be observed that our method achieves very high quality after 7000 epochs (approximately 10 minutes), demonstrating that our approach is highly efficient for reconstructing 4D dynamic scenes. Fig.12 demonstrates that our method can effectively segment dynamic points. Tab. 5 presents the results of several static and dynamic methods on the basketball court datasetVRU (2024), showing that our method outperforms 4DGSWu et al. (2024). To demonstrate the robustness and generalization of our approach, we also conducted experiments on the ENeRF dataset. The results, shown in Table 6, follow the training policies described in 4k4d Xu et al. (2024).

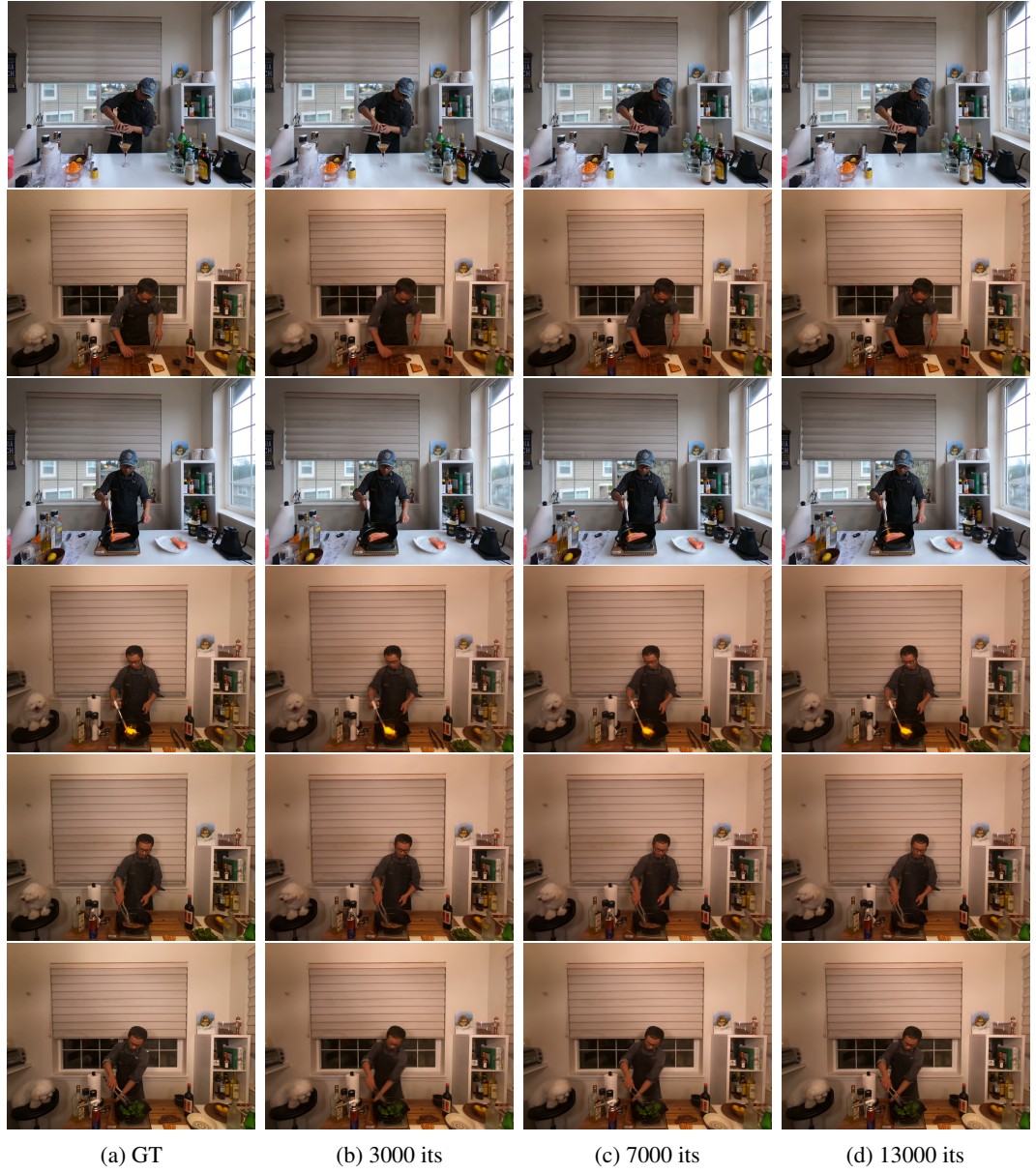

|        (a) GT        |      (b) 3000 its      |      (c) 7000 its      |     (d) 13000 its     |

Figure 11: **Training Epoch Comparison:** the results of our method in 3000, 7000, 13000 epochs. Based on the results from 3000 iterations, our method demonstrates rapid convergence.

Table 5: **Quantitative comparison on the Basketball court dataset.** The first four methods correspond to static methods, tested on the first frame, while the last two methods represent dynamic methods, tested on 20 frames.

| Method | PSNR ↑ | SSIM ↑↓ | LPIPS ↓ |
|---|---|---|---|
| Gof Yu et al. (2024b) | 30.39 | 0.949 | 0.141 |
| 2DGS Huang et al. (2024a) | 30.78 | 0.949 | 0.187 |
| PixelGS Zhang et al. (2024) | 29.26 | 0.946 | 0.168 |
| 3DGSKerbl et al. (2023) | 30.50 | 0.949 | 0.171 |
| 4DGSWu et al. (2024) | 27.87 | 0.921 | 0.191 |
| **Swift4D(Ours)** | 29.03 | 0.933 | 0.187 |

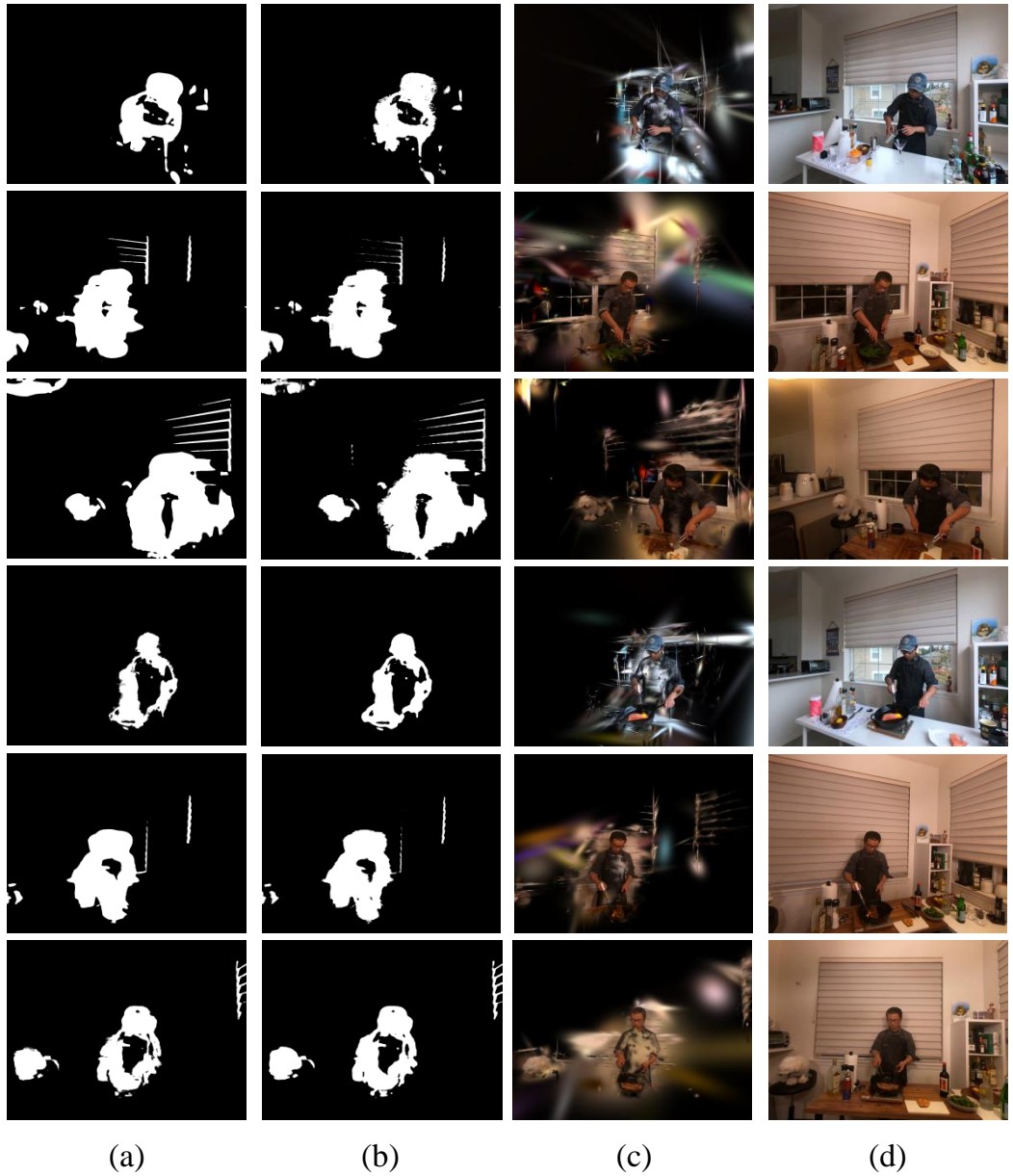

|(a)|(b)|(c)|(d)|

Figure 12: **Decomposition results**. (a) is the dynamic-static pixel mask, (b) is the dynamic map rendered with our dynamic value $d$ of Gaussians, (c) is the image rendered with dynamic Gaussians, and (d) is the GT image.

Table 6: Performance comparison of different methods on ENeRF dataset. The results are derived from 4k4d.

| Methods | PSNR ↑ | SSIM ↑ | LPIPS ↓ |
|---|---|---|---|
| ENeRF Lin et al. (2022) | 25.452 | 0.809 | 0.273 |
| IBRNet Wang et al. (2021) | 24.966 | 0.929 | 0.172 |
| KPlanes Fridovich-Keil et al. (2023) | 21.310 | 0.735 | 0.454 |
| 4k4d Xu et al. (2024) | 25.815 | 0.898 | 0.147 |
| Swift4D (Ours) | 26.12 | 0.911 | 0.070 |

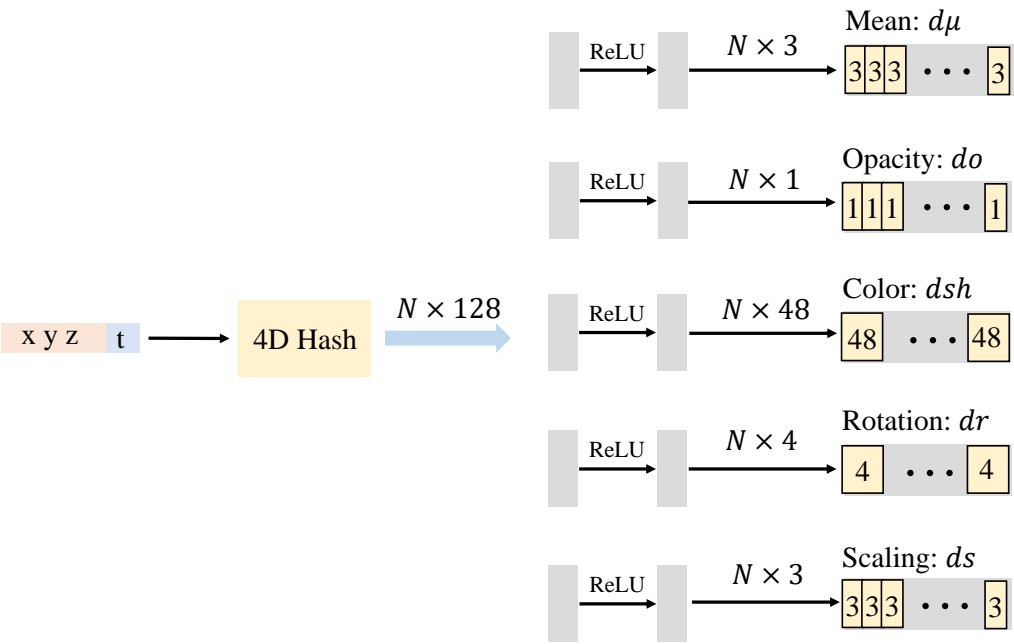

Figure 13: **MLP Structures.** For each dynamic point, we use five small MLPs to predict the deformations.

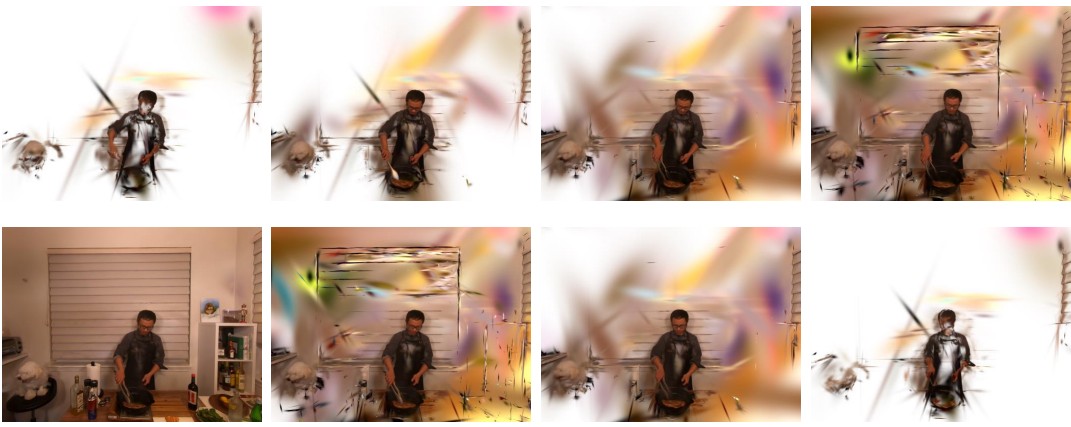

Figure 14: The result of hierarchical rendering in dynamic scenes based on the dynamic parameter $d$ of Gaussian points. The specific video can be found in the supplementary material.

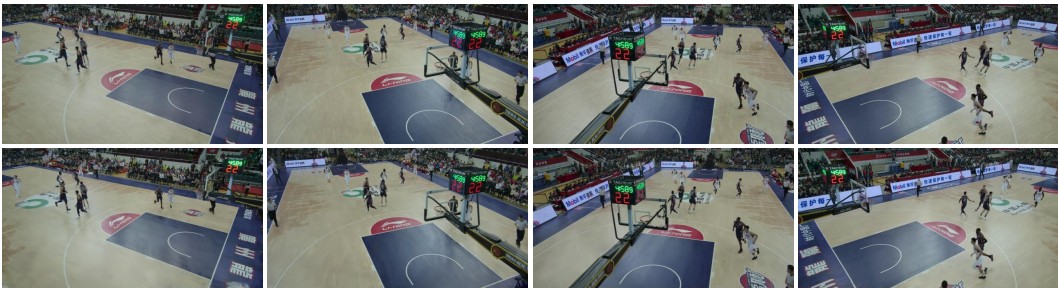

Figure 15: The training results of the basketball court from four novel viewpoints. The images above are the GT images, and the ones below are our rendered results.

