# OpenReview forum: "Swift4D: Adaptive divide-and-conquer Gaussian Splatting for compact and efficient reconstruction of dynamic scene"
_ICLR.cc/2025/Conference — ICLR 2025 Poster_

### Official Review · Reviewer_BxBK · 2024-11-01

**Soundness:** 4
**Presentation:** 3
**Contribution:** 2
**Rating:** 6
**Confidence:** 4

**Summary:**

This paper presents a Gaussian-splatting-based approach for dynamic scene reconstruction. It first builds a canonical space and decomposes the scene into static and dynamic components. Swift4D leverages a compact hash grid to model the deformations within the dynamic components. The proposed method includes a mechanism for identifying dynamic Gaussians and controlling density. Experimental results demonstrate that this method outperforms others.

**Strengths:**

The authors present an innovative approach to decomposing scenes into static and dynamic components by introducing a novel Gaussian attribute combined with alpha blending. To the best of my knowledge, this is the first method to leverage Gaussian splatting for scene decomposition. The results demonstrate its effectiveness on the datasets used, achieving high-quality rendering.

**Weaknesses:**

The primary concern regarding this paper is the lack of sufficient comparisons. While the authors assert that their method is suitable for dynamic scenes, the datasets utilized predominantly feature forward-facing scenes with limited movements. Although the VRU dataset is a multi-view dataset that captures inward-looking scenes, the authors did not provide quantitative comparisons involving it. Additionally, there are other multi-view datasets [1, 2] on which Swift4D could be evaluated, particularly D-NeRF [1], which has been employed in various deformation-based dynamic neural rendering methods that could serve as points of comparison.

The authors mention other methods, Liang et al. (2023) and He et al. (2024), that are also doing static-dynamic decomposition. However, it would be more beneficial to compare their methodologies with the proposed method through quantitative experiments rather than relying solely on qualitative descriptions.

Furthermore, the video results from the VRU dataset exhibit noticeable abrupt scene changes, likely due to the segmentation processing of the video. The authors should address this phenomenon in the paper.


[1] Pumarola, A., Corona, E., Pons-Moll, G., & Moreno-Noguer, F. (2021). D-nerf: Neural radiance fields for dynamic scenes. In Proceedings of the IEEE/CVF Conference on Computer Vision and Pattern Recognition (pp. 10318-10327).
[2] Xu, Z., Peng, S., Lin, H., He, G., Sun, J., Shen, Y., ... & Zhou, X. (2024). 4k4d: Real-time 4d view synthesis at 4k resolution. In Proceedings of the IEEE/CVF Conference on Computer Vision and Pattern Recognition (pp. 20029-20040).

**Questions:**

This paper has some merits. However, the scope and limitations of the proposed method are not clearly defined.

---

> ### Author Response · Authors · 2024-11-17
>
> Thank you for your valuable review comments. I will address each of your concerns here. If there is anything I haven't explained clearly, please don't hesitate to let me know.
>
> **Weaknesses:**
>
> **W1: The primary concern regarding this paper is the lack of sufficient comparisons.**
>
> We fully understand the reviewers' concerns. Following previous multi-view methods [3,4,5,15,16], we use the widely adopted N3DV[4] and Meetroom[5] datasets, which serve as primary benchmarks for current multi-view approaches [2,3,5,15,16]. To make our results more convincing and to demonstrate the generalization capability of our method, we also conducted experiments on the more challenging VRU dataset [17].
> The methods we compare include recent state-of-the-art methods: SpacetimeGS[15], realtimegs[16], 3DGStream[3], 4DGS[10], and others.
>
> In our paper, we include as many datasets and comparison methods as possible. Additionally, we have conducted experiments on the dataset you suggested, as detailed later (**W3**).
>
>
>
>
> **W2: While the authors assert that their method is suitable for dynamic scenes, the datasets utilized predominantly feature forward-facing scenes with limited movements. The authors do not provide test results on the VRU dataset.**
>
> In fact, most multi-view dynamic scene reconstruction methods conduct experiments on these two datasets [4,5]. We follow their choice of datasets to enable direct comparison. To further demonstrate the credibility of our method, we also conduct experiments on the more challenging VRU dataset[17], which features large movements. The quantitative results are provided in the appendix (Line 798) and are as follows:
>
> | Method                      | PSNR ↑ | SSIM ↑ | LPIPS ↓ |
> |-----------------------------|--------|--------|---------|
> | 4DGS [10]       | 27.87  | 0.921  | 0.191   |
> | Swift4D (Ours)              | 29.03  | 0.933  | 0.187   |
>
>
> **W3: There are other multi-view datasets [1, 2], particularly D-NeRF [1], on which Swift4D could be evaluated.**
>
> These two papers include a total of five datasets: D-NeRF [1], DNR-rendering [6], NHR [7], N3DV [8], and ENeRF [9].
>
>
> N3DV [8] is a mainstream multi-view testing dataset, and we have conducted experiments on it, as shown in our paper (line 324). As for ENeRF dataset [9], we follow the 4k4d training policy. Our results are as follows:
>
> | Methods  | PSNR ↑   | SSIM ↑    | LPIPS ↓       |
> |--|--|----|----|
> | ENeRF [9]    | 25.452   | 0.809     | 0.273          |
> | IBRNet [11]   | 24.966   | **0.929**     | 0.172          |
> | KPlanes [13]  | 21.310   | 0.735     | 0.454          |
> | 4k4d [2]    | 25.815   | 0.898     | 0.147          |
> | Swift4D(Ours)     | **26.120**   | 0.911     | **0.070 (alax)**   |
>
>
> As for D-NeRF [1], it is actually a monocular dataset, which is contrary to our multi-view task. DNR-rendering [6] and NHR [7] primarily focus on the reconstruction of dynamic human details, whereas we focus on the reconstruction of dynamic scenes. Humans and scenes represent two distinct tasks, so we do not conduct experiments on their datasets.
>
>
> **W4: It is beneficial to compare the proposed method with those of Liang[13] and He[14].**
>
> Thank you for your suggestion. We will make further qualitative comparisons in the paper.
>
> Here, we also provide a brief introduction: Liang [13] focuses on monocular dynamic scenes, and we did not find multi-view results in their work. Additionally, since their code is not open-sourced, it is difficult for us to make a comparison. He [14] achieves a PSNR of 31.23 dB on the mainstream N3DV dataset, while ours is 32.23 dB.
>
>
>
> **W5:  The video results from the VRU [17] dataset exhibit noticeable abrupt scene changes.**
>
> The scene abrupt changes are caused by the way the basketball video was constructed, as it is made up of four 20-frame video clips stitched together. These abrupt changes occur at the transitions between each clip.
>
> Additionally, the VRU dataset is more closely aligned with real-world scenarios and presents significant challenges. Existing dynamic methods often suffer from severe flickering issues, whereas our approach exhibits minimal flickering and represents the scene more faithfully [3,10]. For further comparison, we have included results of other methods trained on the VRU dataset in the newly added supplementary videos.
>
>
> **Questions:**
>
> **Q1: The scope and limitations of the proposed method are not clearly defined.**
>
> Thank you for the reminder. Dynamic scene reconstruction can be divided into multi-view and monocular dynamic scene reconstruction, with each method presenting different challenges. The main limitation of our method is that, similar to previous work [2,3,5], it focuses on multi-view scenes and currently does not support monocular datasets for dynamic scene reconstruction. Additionally, our method focuses on scene reconstruction and does not include human reconstruction [6,7].

---

> ### Author Response · Authors · 2024-11-18
> **Reference**
>
> **Others:**
>
> We have provided VRU basketball dataset [17] training results from other methods in the Supplementary Material, including the online 3DGStream[1] and offline 4DGS[2]. We sincerely hope you will take the time to review them.
>
> Finally, we sincerely thank you for your questions. Your insights have indeed helped us further refine our paper.
>
> **Reference**
>
> [1] D-nerf: Neural radiance fields for dynamic scenes. CVPR 2021
>
> [2] 4k4d: Real-time 4d view synthesis at 4k resolution. CVPR 2024.
>
> [3] 3DGStream: On-the-fly Training of 3D Gaussians for Efficient Streaming of Photo-Realistic Free-Viewpoint Videos. CVPR 2024
>
> [4] Neural 3d video synthesis from multi-view video. CVPR 2022.
>
> [5] Streaming radiance fields for 3d video synthesis. NIPS 2022.
>
> [6] DNA-Rendering. ICCV 2023.
>
> [7] Multi-view Neural Human Rendering. CVPR 2020.
>
> [8] Neural 3d video synthesis from multi-view video. CVPR 2022.
>
> [9] ENeRF: Efficient Neural Radiance Fields for Interactive Free-viewpoint Video. SIGGRAPH Asia 2022
>
> [10] 4D Gaussian Splatting for Real-Time Dynamic Scene Rendering. CVPR 2024.
>
> [11] IBRNet: Learning Multi-View Image-Based Rendering. CVPR 2021.
>
> [12] K-Planes: Explicit Radiance Fields in Space, Time, and Appearance. CVPR 2023.
>
> [13] Gaufre:Gaussian deformation fields for real-time dynamic novel view synthesis. WACV 2025.
>
> [14] S4D: Streaming 4D Real-World Reconstruction with Gaussians and 3D Control Points. Arxiv.
>
> [15] Spacetime Gaussian Feature Splatting for Real-Time Dynamic View Synthesis. CVPR 2024.
>
> [16] REAL-TIME PHOTOREALISTIC DYNAMIC SCENE REPRESENTATION AND RENDERING WITH 4D GAUSSIAN SPLATTING. ICLR 2024.
>
> [17] VRU. https://anonymous.4open.science/r/vru-sequence/. 2024.

---

> ### Author Response · Authors · 2024-11-23
>
> Apologies for the interruption. We sincerely hope you will continue to engage in the discussion. If you have any further questions or concerns, we are more than willing to provide additional explanations or supporting materials. Your insights are crucial for refining our research and ensuring its relevance and impact. Thank you very much.

---

> ### Author Response · Authors · 2024-11-27
>
> Dear Reviewer BxBK,
>
> We sincerely thank you for your valuable feedback during the first round of review and for the thoughtful discussions that have greatly contributed to improving our work. Your insights and suggestions have been instrumental in refining our submission, and we are deeply grateful for your time and effort.
>
> We kindly wish to confirm whether we have satisfactorily addressed all your concerns. Thank you again for your devotion to the review. If all the concerns have been successfully addressed, please consider raising the scores after this discussion phase.
>
> Best regards,
>
> Paper4181 Authors

---

> ### Comment · Reviewer_BxBK · 2024-11-27
>
> I appreciate the authors' feedback. It has addressed most of my concerns. Thus, I will raise the score.

---

> > ### Author Response · Authors · 2024-11-27
> >
> > Dear Reviewer BxBK:
> >
> > Thank you for your response, valuable comments, and recognition of our work. This has helped make our work more complete.
> >
> > Best Regards,
> >
> > Paper 4181 Authors

---

### Official Review · Reviewer_c4Zw · 2024-11-02

**Soundness:** 3
**Presentation:** 3
**Contribution:** 3
**Rating:** 8
**Confidence:** 5

**Summary:**

The paper introduces a 4D Gaussian representation for dynamic scenes, utilizing a learned deformation field to model Gaussian dynamics. To reduce the redundancy in dynamic learning, the method employs a decomposition strategy to separate dynamic and static elements. For the dynamic components, 4D spatiotemporal coordinates query a 4D hash grid, retrieving features from multi-resolution grids. An MLP then predicts the residual values of Gaussian attributes to capture the dynamics. By isolating dynamic from static components, this streamlined model efficiently handles dynamic modeling while occupying less space.

**Strengths:**

1. Decomposing static and dynamic elements is beneficial, effectively reducing the model's learning burden and increasing inference speed, as shown in Table 3. Notably, this separation is also present in the concurrent work GauFRe (Liang et al.) and the published work SC-GS (Huang et al.). This concept is a major contribution to GauFRe and a detailed implementation in SC-GS. Citing these methods when discussing this idea is appropriate.

2. The pruning strategy in Section 3.4 is innovative. Gaussian Splatting and NeRF, based on Multi-View Stereo (MVS) principles, converge when multi-view images reach a photometric consensus. An isolated Gaussian point, recognized by only a few views, tends to become a floating artifact, causing overfitting. This insightful point is valuable and inspires the community.

The two points already make me inclined to accept the paper, but there are some concerns. Please refer to the weaknesses section.

**Weaknesses:**

The experimental results and citations are insufficient. The FPS of Swift4DLite (Ours) is missing in Table 1, with no explanation provided. Additionally, the rendering speed is relatively slow, similar to 4DGS (Yang et al.) and SpaceTimeGS, with comparable rendering quality. The training time for SpaceTimeGS reported in Table 1 is too long; it should be under an hour. The FPS of SpaceTimeGS is also lower than I expected (130+). Moreover, several closely related works are not compared or cited:

1. Huang, Yi-Hua, et al. "SC-GS: Sparse-controlled Gaussian splatting for editable dynamic scenes." Proceedings of the IEEE/CVF Conference on Computer Vision and Pattern Recognition. 2024.
2. Yang, Ziyi, et al. "Deformable 3D Gaussians for high-fidelity monocular dynamic scene reconstruction." Proceedings of the IEEE/CVF Conference on Computer Vision and Pattern Recognition. 2024.
3. Lin, Youtian, et al. "Gaussian-flow: 4D reconstruction with dynamic 3D Gaussian particles." Proceedings of the IEEE/CVF Conference on Computer Vision and Pattern Recognition. 2024.

I recommend at least including comparisons with (1) and (3), as these methods model Gaussian dynamics using spatially and frequency-compact bases, respectively. Additionally, SC-GS achieves high FPS in rendering images by inferring the motion of sparse control points, according to their Supp.

**Questions:**

I hope the author can respond to my concerns and suggestions mentioned in the weaknesses section. If there are any issues with the experimental results, recommended citations, or comparisons, I would appreciate an explanation.

---

> ### Author Response · Authors · 2024-11-18
>
> Thank you very much for your recognition. We will address your concerns point by point and revise the paper based on your suggestions.
>
> **W1:  The FPS of Swift4DLite (Ours) is missing in Table 1, with no explanation provided.**
>
> The missing FPS for Swift4DLite is omitted because it is close to that of Swift4D. We will modify the paper accordingly.
>
> **W2: The rendering speed is relatively slow, similar to 4DGS (Yang et al.) and SpaceTimeGS, with comparable rendering quality.**
>
> Yes, our rendering speed is indeed similar to theirs, but our method actually achieves 125 FPS on dynamic scene datasets (Resolution: 1352x1014), which, to the best of our knowledge, is currently the highest performance.
>
> **W3: The training time of SpaceTimeGS reported in Table 1 is too long; it should be under an hour. The FPS of SpaceTimeGS is also lower than my expectations (130+).**
>
> Actually, the training time reported in their paper is based on training 50 frames on an **RTX A6000** (as shown in Section 5: Implementation Details of their paper). However, during our training on the **NVIDIA RTX 3090**, we find that their reported training time for 50 frames is around 1.5-2 hours. Therefore, their total training time for 300 frames would be approximately 10 hours. As for FPS, we faithfully report the results from our NVIDIA RTX 3090 tests. The difference between our results and those reported in their paper may be due to the difference in GPU models.
>
>
> **W4: Several closely related works have not been compared or cited.**
>
> SC-GS [1] is indeed a work closely related to ours and achieves very high FPS and rendering quality. However, they [1,2] seem to focus more on monocular dynamic scene reconstruction, while we focus on multi-view dynamic scene reconstruction. Therefore, comparing our work with these two studies presents certain challenges. We will also explain this point in the paper.
>
> As for [3], we will report their results in the paper. Their results on the mainstream multi-view dataset N3DV [6] are as follows.
>
> method | PSNR | time
> -|-|-
> Gaussian-Flow[3] | 32.00 | 41.8 min
> Swift4D(Ours) | 32.23 | 25.0 min
>
> **Others:**
>
> We have provided VRU basketball training results from other methods in the Supplementary Material, including the online 3DGStream[1] and offline 4DGS[2]. We sincerely hope you will take the time to review them.
>
> Finally, we sincerely thank you for your questions. Your insights have indeed helped us further refine our paper.
>
> **Reference**
>
> [1] SC-GS: Sparse-controlled Gaussian splatting for editable dynamic scenes. CVPR 2024.
>
> [2] Deformable 3D Gaussians for high-fidelity monocular dynamic scene reconstruction. CVPR 2024.
>
> [3] Gaussian-flow: 4D reconstruction with dynamic 3D Gaussian particles. CVPR 2024.
>
> [4] 3DGStream: On-the-fly Training of 3D Gaussians for Efficient Streaming of Photo-Realistic Free-Viewpoint Videos. CVPR 2024
>
> [5] Real-time Photorealistic Dynamic Scene Representation and Rendering with 4D Gaussian Splatting. ICLR 2024.
>
> [6] Neural 3d video synthesis from multi-view video. CVPR 2022.

---

> > ### Comment · Reviewer_c4Zw · 2024-11-19
> >
> > Thank you to the authors for the feedback. The response addresses my concerns. I also have a recommendation, as noted by R-9PWz, to further explore the differences in the pruning strategy compared to existing methods. This is one of the most inspiring aspects of the paper, and I hope the code will be published to benefit the community. I have no further concerns and will raise my score, allowing the authors to focus on feedback from other reviewers.

---

> > > ### Author Response · Authors · 2024-11-19
> > >
> > > Thank you for your recognition and understanding. We will carefully refine the paper based on the feedback from all reviewers and will also open-source the code and model.

---

### Official Review · Reviewer_TAqw · 2024-11-02

**Soundness:** 3
**Presentation:** 3
**Contribution:** 2
**Rating:** 6
**Confidence:** 4

**Summary:**

This paper proposes Swift4D, which extends 3DGStream with a static & dynamic decoupling strategy. Doing this significantly reduced the model size and improved the rendering quality. The authors also introduce other tricks like  spatio-temporal structure modeling and adjusted density control.

**Strengths:**

* The motivation and technique design for this paper is straightforward and reasonable. Decoupling static and dynamic contents is an intuitive approach when dealing with real dynamic scenes. Although the approach is questionable, it may help researchers to understand the problem better and leverage the explicit structure of 3DGS.
* The experiment results on different datasets show improvements in various aspects, like training time and PSNR, though the FPS is lower, and this is not an on-the-fly method.
* The writing and illustration of this paper is pretty clear to me.

**Weaknesses:**

* One critical weakness is the implementation of decoupling static and dynamic parts. Specifically, they use 2D supervision to build the dynamic factor. However, the way they build 2D masks is just to calculate the temporal variance of each pixel, which should lead to problems when dynamic movements happen in textureless regions. These regions can easily be recognized as static parts. Such a phenomenon could be observed in the sear_steak_static result of the supplementary, where part of the cloth is classified as static. In addition, calculating the variance throughout the entire time could cause problems in long-sequence video.
* It seems that this work can't deal with single-view videos like DNeRF. This could be a general limitation as it's hard to deploy and compare with other methods that can deal with such cases (e.g., SCGS). In addition, it needs to be trained offline, making the comparison with 3DGSStream kind of unfair.
* To me, the result in Fig.7 does actually demonstrate the sensitivity of the dynamic threshold, where it affects the PSNR a lot. I am worried about the generalizability of the method since it's tested on scenes with similar dynamic-static compositions.

**Questions:**

As described in weakness, my questions mainly focus on the generalizability to texture-less regions, highly dynamic videos (with different static-dynamic compositions), and long-term videos.

---

> ### Author Response · Authors · 2024-11-18
> **W1-W3**
>
> Thank you very much for your recognition and valuable feedback. Here, we will address your concerns point by point. If there are any areas where our explanations are unclear, please feel free to point them out.
>
> **W1: Using the 2D mask calculated from the temporal variance of pixels as supervision may lead to issues when dynamic motion occurs in textureless areas. These regions can easily be recognized as static parts.**
>
>
> We can classify weak-texture dynamic elements into two categories: one is weak-texture dynamic elements with large-scale motion, and the other is weak-texture elements with fine-scale motion.
>
> For the first category, as seen in our supplementary video 1 basketball, the weak-texture clothing worn by the athlete is classified as a dynamic element and deforms along with the athlete's movement. Therefore, our method performs well in such situations.
>
> For the second category, using 2D masks as supervision can indeed lead to some weak-texture dynamic elements with limited motion being classified as static, such as the central area of fabric.
> However, this issue only occurs in the central area of textureless item (fabric), while the dynamic classification still holds for the edges of these elements, as seen in the supplementary video sear_steak_dynamic. In fact, because the Gaussian points in the central region of weak-texture objects have the same color, even with fine-grained motion, their final rendering results are visually the same as those in a static state. Therefore, it is reasonable to classify these Gaussian points as either dynamic or static.
>
> Therefore, using the 2D mask calculated from the temporal variance of pixels as supervision is reasonable.
>
> Additionally, the choice of supervision signal is flexible; it can be the 2D mask calculated from the temporal variance of pixels, SAM [1], or other methods. We simply provide a basic approach using variance as supervision.
>
>
>
> **W2: Calculating the variance throughout the entire time could cause problems in long-sequence video.**
>
> We are not clear on the specific problems you are referring to. Please allow us to respond based on our interpretation of the issue.
>
> Firstly, current dynamic methods do not address the challenge of dynamic modeling for long-sequence videos. Directly performing dynamic reconstruction on long-sequence videos is uncommon, as modeling the dynamic information of an entire long-sequence video with a single model is highly challenging (a common issue with current offline methods [2,5,6,7,8]). This often results in severe blurring and flickering artifacts. **A common and practical solution is to divide such videos into shorter sequences (approximately 300 frames) and perform dynamic reconstruction on each sequence separately.** For a 300-frame sequence, our variance computation approach functions without any issues.
>
> Secondly, even in the case of long-sequence videos requiring variance computation, the video can be divided into smaller batches, with variance computed separately for each batch. The overall variance for the long-sequence video can then be derived using the variance transformation formula.
>
>
> **W3:  It seems that this work can't deal with single-view videos like DNeRF. This could be a general limitation as it's hard to deploy and compare with other methods that can deal with such cases (e.g., SCGS).**
>
>
> Yes, similar to 3DGStream[4] and SpaceTimeGS[5], our method focuses on multi-view dynamic scene reconstruction and can not  handle single-view videos like D-NeRF. In contrast, SCGS[8] and D-NeRF[6] focus more on monocular datasets and struggle to handle multi-view datasets well [2,3]. All of these methods tackle different tasks, making a direct comparison between them challenging. Both monocular and multi-view dynamic scene reconstruction come with their own challenges.

---

> ### Author Response · Authors · 2024-11-18
> **W4-W6**
>
> **W4: In addition, it needs to be trained offline, making the comparison with 3DGSStream kind of unfair.**
>
> Yes, our method is trained offline, and we will specifically label 3DGStream as an online training method.
>
>
> **W5: To me, the result in Fig.7 does actually demonstrate the sensitivity of the dynamic threshold, where it affects the PSNR a lot.**
>
> Considering reviewer's concern, the PSNR could fluctuate slightly with small adjustments in the threshold due to the smaller size of the N3DV datasets[2]. Therefore, we conducted ablation experiments on the VRU basketball court dataset[9], which has a larger dynamic range and coverage. The experimental results are as follows:
>
> | Dynamic Threshold | PSNR   | Dynamic-Static Compositions |
> |--------------------|--------|----------------------------|
> | 3.0               | **28.76**  | 0.230                      |
> | 4.0               | **28.89**  | 0.215                      |
> | 5.0               | **28.82**  | 0.204                      |
> | 6.0               | **28.81**  | 0.187                      |
> | 7.0               | **29.10**  | 0.173                      |
> | 8.0               | **28.62**  | 0.163                      |
> | 9.0               | **28.57**  | 0.138                      |
>
> The impact of the dynamic threshold hyperparameter on PSNR is expected. As the dynamic threshold increases, fewer Gaussian points are classified as dynamic, making dynamic scene modeling more challenging and resulting in decreased PSNR. In our paper, **we consistently set the dynamic threshold to 7.0 for all experiments**, achieving excellent results, which demonstrates the robustness and generalizability of this threshold across various scenarios.
>
> **W6: Worried about the generalizability of the method since it's tested on scenes with similar dynamic-static compositions.**
>
>
> In fact, our dynamic-static separation approach is specifically designed for scenarios where the static portion is large  and the dynamic portion is small. This dynamic-static composition closely reflects real-world scenes, and extensive experiments prove that our method is very effective.
>
> If a scene, as pointed out by the reviewer, has a different dynamic-static composition (**with the dynamic proportion being large, 70% or even higher**), then applying a dynamic-static separation strategy may not be necessary. Directly modeling the entire scene might be more reasonable in such cases. In such cases, applying our dynamic-static separation approach may not yield significant benefits, as the performance gains from reducing the computational load on the static part are limited when the dynamic portion is large.
>
> Considering the reviewer’s concerns, we tested our method on the VRU dataset, where the dynamic-static compositions exhibit varying distributions (ranging from 0.138 to 0.230) based on adjustments to the dynamic threshold. The experimental results are as follows:
>
>
> | Dynamic Threshold | PSNR   | Dynamic-Static Compositions |
> |--------------------|--------|----------------------------|
> | 3.0               | 28.76  | **0.230**                      |
> | 4.0               | 28.89  | **0.215**                      |
> | 5.0               | 28.82  | **0.204**                      |
> | 6.0               | 28.81  | **0.187**                      |
> | 7.0               | 29.10  | **0.173**                      |
> | 8.0               | 28.62  | **0.163**                      |
> | 9.0               | 28.57  | **0.138**                      |
>
>
> **Others:**
>
> We have provided training results from other methods in the Supplementary Material, including the online 3DGStream[1] and offline 4DGS[2]. We sincerely hope you will take the time to review them.
>
> Finally, we sincerely thank you for your questions. Your insights have indeed helped us further refine our paper.
>
>
>
>
> **Reference**
>
> [1] Segment Anything. ICCV 2023.
>
> [2] Neural 3d video synthesis from multi-view video. CVPR 2022.
>
> [3] Streaming radiance fields for 3d video synthesis. NIPS 2022.
>
> [4] 3dgstream: On-the-fly training of 3d gaussians for efficient streaming of photo-realistic free-viewpoint videos
>
> [5] Spacetime Gaussian Feature Splatting for Real-Time Dynamic View Synthesis. CVPR 2024.
>
> [6] D-NeRF: Neural Radiance Fields for Dynamic Scenes. CVPR 2021.
>
> [7] Deformable 3D Gaussians for High-Fidelity Monocular Dynamic Scene Reconstruction. CVPR 2024.
>
> [8] SC-GS: Sparse-controlled Gaussian splatting for editable dynamic scenes. CVPR 2024.
>
> [9] VRU. https://anonymous.4open.science/r/vru-sequence/. 2024.

---

> ### Author Response · Authors · 2024-11-23
>
> Apologies for the interruption. We sincerely hope you will continue to engage in the discussion. If you have any further questions or concerns, we are more than willing to provide additional explanations or supporting materials. Your insights are crucial for refining our research and ensuring its relevance and impact. Thank you very much.

---

> > ### Comment · Reviewer_TAqw · 2024-11-26
> >
> > I appreciate the effort made by the authors, and most of my concerns are addressed. Although I still worry about robustness since it has certain assumptions about the 4D scene, like the potential ratio of static and dynamic parts, it seems challenging to propose a general solution in this field.  I will keep my current positive rating.

---

> ### Author Response · Authors · 2024-11-26
>
> Thank you for your feedback and understanding. As you pointed out, proposing a universal and effective method in this field is indeed challenging. We would also like to clarify that we do not explicitly assume a fixed ratio of static to dynamic parts in a scene. Our testing data revealed that static backgrounds generally dominate, while dynamic components are relatively minor. In fact, this is a common characteristic of real-world scenes, which has been validated and adopted by several dynamic-static separation methods, such as [MSTH, GauFRe]. Therefore, **treating all points (whether static or dynamic) as dynamic** and applying temporal modeling to all of them— as current dynamic modeling methods do— is inefficient and unreasonable, leading to significant unnecessary computational overhead. Based on this observation, we have established the theoretical foundation for a dynamic-static separation strategy, which excludes static points from dynamic modeling, **enabling efficient modeling of real dynamic scenes.**
>
> Specifically, in dynamic scenes, the higher the proportion of static parts, the more pronounced the advantages of our method compared to other dynamic modeling approaches, as **we avoid performing dynamic modeling on static components**. In scenarios with a very high proportion of dynamic components, our method may perform similarly to other approaches like [3DGstream, 4DGaussian, SpacetimeGS]. However, such scenarios are rare and typically designed for specific tasks. For instance, in human-centric datasets like [DNR-Rendering, NHR], which consist only of dynamic foregrounds, the advantages of our method may be less apparent, as modeling all components dynamically is reasonable for such specialized datasets.
>
> To date, our dynamic-static separation method has demonstrated high efficiency across various datasets, including mainstream datasets like **N3DV** and **MeetRoom**, the more challenging **VRU basketball dataset**, and the outdoor **E-NeRF dataset**. These results highlight the robustness of our method, its adaptability to diverse real-world scenarios, and its significant potential for practical applications.
>
> The ENeRF dataset test results are as follows:
>
> | Methods  | PSNR ↑   | SSIM ↑    | LPIPS ↓       |
> |--|--|----|----|
> | ENeRF [9]    | 25.452   | 0.809     | 0.273          |
> | IBRNet [11]   | 24.966   | **0.929**     | 0.172          |
> | KPlanes [13]  | 21.310   | 0.735     | 0.454          |
> | 4k4d [2]    | 25.815   | 0.898     | 0.147          |
> | Swift4D(Ours)     | **26.120**   | 0.911     | **0.070 (alax)**   |
>
> Thank you very much for your response. We look forward to further discussions with you.

---

### Official Review · Reviewer_9PWz · 2024-11-04

**Soundness:** 2
**Presentation:** 3
**Contribution:** 2
**Rating:** 6
**Confidence:** 4

**Summary:**

The sources describe Swift4D, a novel method for reconstructing dynamic scenes by separating static and dynamic elements and using a 4D hash grids to model the temporal information of dynamic Gaussian points. The separation improves efficiency, reducing both storage requirements and training time, while achieving strong rendering quality in various dynamic scene scenarios. The authors test their method on  N3DV, MeetRoom and Basketball court datasets and compare to different recent works.

**Strengths:**

- The paper is overall well written and easy to understand. The authors provide detailed descriptions of the methodology and experiments.
- According to the quantitative evaluation, the method performs favourably to competing approaches in image quality as well as optimization time.
- The idea is relatively simple of static and dynamic decomposition, but appears to be effective in terms of speed.

**Weaknesses:**

- A key component of this work is the dynamic and static decomposition. The method introduces a dynamic level parameter that is optimized for each gaussian. As explained in section 3.2, a 2D dynamic-static pixel Mask is computed from the training video by leveraging the temporal standard deviation of each pixel. However, since the video can have camera motion as well as dynamic elements that yield varying the pixel intensity it is unclear to me how the standard deviation can serve as good criterion to  differentiate dynamic scene elements from static ones. Does this criterion work well on videos with strong camera motion?
- The authors propose a temporal pruning strategy, it would be good to discuss at some point existing pruning strategies, e.g. LightGaussians (Fan et al. NeurIPS 2024), and the relation ship the proposed pruning method.
- In the supplementary material videos, there seem to be artifacts of vanishing dynamic elements. In 2_baskball2.mp4 some players vanishing from frame to frame and it appears to have temporal inconsistencies, that questions the effectiveness of the method. Is this a common issue?

**Questions:**

- Please comment on the  Weaknesses above.
- How can the model differentiate between specular reflections and dynamic elements?

**Details Of Ethics Concerns:**

No concerns

---

> ### Author Response · Authors · 2024-11-18
>
> Thank you for your review. We will make improvements based on your feedback. Next, we will address your concerns point by point. If there are any areas where our explanations are unclear, please feel free to point them out.
>
> **W1: However, since the video can have camera motion as well as dynamic elements that yield varying the pixel intensity it is unclear to me how the standard deviation can serve as good criterion to differentiate dynamic scene elements from static ones.**
>
> Dynamic scene reconstruction methods are typically categorized into two approaches: monocular video input (involving camera motion, e.g., SCGS [10], DeformableGS [9]) and multi-view video input (excluding camera motion, e.g., 3DGStream [1], SpaceTimeGS [3]), each presenting distinct challenges.
>
> The "videos with camera motion" you mentioned may fall into the category of monocular videos. However, similar to previous works [1, 3], our method focuses on multi-view scene reconstruction, where the cameras are fixed. Therefore, the concern you raised about "camera motion causing...static elements" has not been considered in our approach, as it primarily targets multi-view scenes where camera motion is absent.
>
> Standard deviation is indeed unsuitable for monocular dynamic scenes. However, in multi-view dynamic scenes, it can serve as an effective supervisory signal. To validate the effectiveness of our method, we conducted extensive experiments across various datasets, including mainstream benchmarks like **N3DV** and **MeetRoom**, as well as more challenging scenarios such as large-motion scenes in **VRU Basketball** and outdoor environments in **ENeRF**.
>
>
> **W2: Does this criterion work well on videos with strong camera motion?**
>
> Aligned with previous works (3DGStream [1], SpaceTime [3]), a limitation of our method, as well as theirs, is its focus on multi-view dynamic scene reconstruction ( without camera motion), making it unsuitable for reconstructing dynamic scenes from monocular videos ( with strong camera motion ). Consequently, our method is not applicable to monocular dynamic datasets with camera motion.
>
> **W3: Discuss the differences between our approach and existing pruning strategies.**
>
> Thank you for your suggestions. We will elaborate on the differences in the relevant section and implement the changes in the PDF file as outlined below.
>
> **LightGS[8] prunes Gaussian points in the spatial domain based on attributes such as opacity and volume. In contrast, our method focuses on pruning Gaussian points based on their importance across all training views in the entire temporal domain.**
>
>
> **W4:  there seem to be artifacts of vanishing dynamic elements. In 2_baskball2.mp4 some players vanishing from frame to frame and it appears to have temporal inconsistencies.**
>
> **Temporal inconsistencies**: The basketball video we provided is composed of several 20-frame video clips stitched together, which may result in frame-to-frame inconsistencies. These inconsistencies are not caused by our method.
>
> **Artifacts of disappearing dynamic elements**: We are the first to experiment with such a large-scale dynamic dataset. The main cause of issues like disappearance and flickering is the dataset's increased complexity and larger motion amplitudes, which is common in current dynamic methods. **We have provided VRU training results from other methods in the Supplementary Material, including the online 3DGStream[1] and offline 4DGS[2].**
>
> **W5: How can the model differentiate between specular reflections and dynamic elements?**
>
> Specular reflections are a challenging problem [4,5,6]. While current dynamic methods don't address them, we are open to exploring this issue. Our method currently cannot distinguish specular reflections from dynamic elements. If a moving object is reflected in a mirror, it treats the reflection as dynamic, marking the Gaussians in the mirror region to model the changes.
>
> Finally, we sincerely thank you for your questions. Your insights have indeed helped us further refine our paper.

---

> > ### Author Response · Authors · 2024-11-19
> > **Reference**
> >
> > **Reference**
> >
> > [1] 3DGStream: On-the-fly Training of 3D Gaussians for Efficient Streaming of Photo-Realistic Free-Viewpoint Videos. CVPR 2024
> >
> > [2] 4D Gaussian Splatting for Real-Time Dynamic Scene Rendering. CVPR 2024.
> >
> > [3] Spacetime Gaussian Feature Splatting for Real-Time Dynamic View Synthesis. CVPR 2024.
> >
> > [4] SpecNeRF: Gaussian Directional Encoding for Specular Reflections. CVPR 2024.
> >
> > [5] Spec-Gaussian: Anisotropic View-Dependent Appearance for 3D Gaussian Splatting. NIPS 2024.
> >
> > [6] MirrorGaussian: Reflecting 3D Gaussians for Reconstructing Mirror Reflections. ECCV 2024.
> >
> > [7] Segment Anything. ICCV 2023.
> >
> > [8] LightGaussian: Unbounded 3D Gaussian Compression with 15x Reduction and 200+ FPS. NIPS 2024.
> >
> > [9] Deformable 3D Gaussians for High-Fidelity Monocular Dynamic Scene Reconstruction. CVPR 2024.
> >
> > [10] SC-GS: Sparse-controlled Gaussian splatting for editable dynamic scenes. CVPR 2024.
> >
> > [11] Neural 3d video synthesis from multi-view video. CVPR 2022.
> >
> > [12] Streaming radiance fields for 3d video synthesis. NIPS 2022.
> >
> > [13] VRU. https://anonymous.4open.science/r/vru-sequence/. 2024.

---

> ### Author Response · Authors · 2024-11-23
>
> Apologies for the interruption. We sincerely hope you will continue to engage in the discussion. If you have any further questions or concerns, we are more than willing to provide additional explanations or supporting materials. Your insights are crucial for refining our research and ensuring its relevance and impact. Thank you very much.

---

> ### Author Response · Authors · 2024-11-27
>
> Dear Reviewer 9PWz,
>
> We sincerely thank you for your valuable feedback during the first round of review and for the thoughtful discussions that have greatly contributed to improving our work. Your insights and suggestions have been instrumental in refining our submission, and we are deeply grateful for your time and effort.
>
> We kindly wish to confirm whether we have satisfactorily addressed all your concerns. Thank you again for your devotion to the review. If all the concerns have been successfully addressed, please consider raising the scores after this discussion phase.
>
> Best regards,
>
> Paper4181 Authors

---

> > ### Comment · Reviewer_9PWz · 2024-11-28
> >
> > Thanks for providing clarifications on the Mono Video and Multi-view setup. Please emphasize this in a final version. The pruning discussion looks good to me and the explanation for the temproal inconsistencies and specular effects sound plausible. I appreciate your efforts and raise my score to 6.

---

> > > ### Author Response · Authors · 2024-11-28
> > >
> > > Dear Reviewer 9PWz:
> > >
> > > Thank you for your response, valuable comments, and recognition of our work. We will improve our paper based on your comments.
> > >
> > > Best Regards,
> > >
> > > Paper 4181 Authors

---

### Author Response · Authors · 2024-11-23
**Global Response**

We sincerely thank all four reviewers for their constructive feedback. We greatly appreciate their recognition of our contributions:

1. The innovation of our idea, especially the explicit dynamic-static decomposition method. (Reviewers 9PWz, TAqw, c4Zw, BxBK).
2. Strong experimental results, particularly in training speed (Reviewers 9PWz, TAqw, c4Zw, BxBK).
3. Clear and concise writing (Reviewers 9PWz, TAqw).

In response to the reviewers’ concerns, we have provided detailed, point-by-point replies and incorporated corresponding revisions into the main paper. The updated sections are highlighted in $\textcolor{orange}{orange}$.

---

### Author Response · Authors · 2024-11-25
**Looking forward to discussions.**

Dear reviewers,

We sincerely appreciate the time you dedicated to reviewing our work and the recognition of our efforts. In response to the concerns you raised, we have provided detailed explanations in our replies. We hope that these address your points adequately. If there are any aspects of our work that remain unclear, please do not hesitate to reach out.

Best regards,
Authors

---

### Meta-Review · Area_Chair_Hfyp · 2024-12-21

**Metareview:**

This paper introduces a method for reconstructing dynamic scenes by decoupling static and dynamic elements. This decomposition reduces both storage requirements and training time. The idea of explicitly separating dynamic and static components is novel, and the temporal pruning strategy is also innovative. The proposed approach demonstrates favorable performance compared to competing methods in terms of image quality and training speed. Despite these strengths, concerns remain regarding the robustness and generalizability of the static-dynamic decomposition. Given that the problem is challenging, limitations are acceptable. However, the paper should discuss these limitations more thoroughly.

**Additional Comments On Reviewer Discussion:**

Several issues were raised in the initial reviews, and the rebuttal effectively addressed most of them.

Several concerns arose due to confusion about the problem setting. Since the paper focuses on a multi-view scenario, questions and concerns based on a monocular perspective do not apply. Accordingly, the paper should clarify its problem setting to prevent such misunderstandings.

Some reviewers expressed concerns about the robustness and generalizability of the static and dynamic element decomposition. The rebuttal clarified circumstances where the proposed method might encounter difficulties and provided results on the VRU and ENeRF datasets to mitigate these concerns. While some reservations remained, the reviewers acknowledged that the problem is challenging and that limitations are acceptable. Nonetheless, the paper should include a more thorough discussion of these limitations.

In addition, concerns were raised regarding the lack of sufficient comparisons. Some comparisons were less relevant due to differences in experimental settings. The rebuttal addressed the remaining issues by presenting new experiments on the N3DV and VRU datasets.

The rebuttal effectively addressed most of the concerns raised during the review process. All reviewers were optimistic about the paper by the end of the discussion stage.

---

### Decision · Program_Chairs · 2025-01-22

Accept (Poster)